# Novel Fuzzy Composite Indicators for Locating a Logistics Platform under Sustainability Perspectives

**Hana Ayadi** [1,2,3,]*[iD]**, Nadia Hamani** [2][iD]**, Lyes Kermad** [1][iD] **and Mounir Benaissa** [3]

1   IUT de Montreuil, University of Paris 8, 93100 Montreuil, France; l.kermad@iut.univ-paris8.fr
2   INSSET, University of Picardie Jules Verne, 02100 Saint Quentin, France; nadia.hamani@u-picardie.fr
3   ISGIS, University of Sfax, Sfax 3021, Tunisia; mounir.benaissa@isgis.usf.tn
*   Correspondence: hana.ayedi@etud.univ-paris8.fr

**Abstract:** The purpose of this paper is to help decision-makers choose the location of a logistics platform with sustainability perspectives. This study presents a compensatory and partially compensatory approach to build composite indicators, using mainly fuzzy multi-criteria decision-making methods. In the first instance, the fuzzy full consistency method (F-FUCOM) was used to calculate the weight of the criteria and sub-criteria. In the second instance, two aggregation methods, namely the fuzzy multi-attribute ideal-real comparative analysis (F-MAIRCA) and the fuzzy preference ranking organization method for enrichment evaluation (F-PROMETHEE), were used to rank the location of a logistics platform. The novelty of the work lays in studying the impact of limited sustainability and weak sustainability on the location of a logistics platform. In this respect, the aggregation of various sustainability criterion in fuzzy compensatory and partially compensatory composite indicators is an innovative and interesting approach used to locate a logistics platform. The obtained results show that economic sustainability is the most important criterion for the selection of a logistics platform, followed by the environmental criterion. Obviously, the F-MAIRCA and F-PROMETHEE methods provided the same ranking orders. Finally, sensitivity analyses were performed to validate the robustness of the proposed approach.

**Keywords:** logistics platform; facility location selection; composite indicators; multi-criteria decision-making; F-FUCOM; F-PROMETHEE; F-MAIRCA; sustainability

## 1. Introduction

In an increasingly competitive environment, the freight transport system is undeniably necessary to ensure the proper functioning of the city economy through its presence in the upstream and downstream of the supply chain [1,2]. Despite its importance, this system has not been given great attention by the local, regional and national authorities. Thus, to make the movement of goods more fluid, it is essential to know the characteristics of the urban space, its stakes and its constraints [3,4]. To this end, the establishment of a logistics platform, located a few kilometers from the city center, can certainly alleviate the severity of the impact of freight transport on the city, and in terms of sustainable development, by making deliveries more fluid.

The logistics platform is an infrastructure that affects sustainable development by reducing traffic congestion, minimizing carbon emissions and ensuring efficient land use. The main mission of this platform is to pool resources and to decrease the concentration of flows to the city or, more precisely, to the city center [5]. It also reduces transport costs, delays and nuisances, and facilitates the flow of goods and the transition from one mode of transport to another, using technologically advanced and efficient equipment to handle containers. These activities accompanied by storage operations require large, equipped warehouses.

However, the implications of building infrastructure and operating logistics facilities are far-reaching and often almost imperceptible. The operation of a logistics platform

negatively and positively affects the environment of the zone where it is implemented. On the other hand, there are vertical and horizontal relationships between the interests of the actors associated with the implementation of a logistics platform [6].

In this context, the next section presents the existing work addressing the problem of facility location. Our study of the literature shows that no general or systematic method of localizing a logistics platform, specifically related to the sustainability perspectives, has been proposed until now. In fact, this issue depends on a set of locations (alternatives) assessed against a set of weighted criteria that are independent of each other. In this research work, an innovative and interesting approach of locating a logistics platform, based on fuzzy compensatory and partially compensatory composite indicators, is introduced as a support tool for decision-makers to locate a logistics platform with sustainability perspectives. The following research questions are answered in this paper: What are the important criteria that should be applied to select a logistics platform in order to increase the level of sustainability? What are the most commonly used multi-criteria decision-making (MCDM) methods for weighting and aggregation as reported in the literature? How can they be utilized well in fuzzy compensatory and partially compensatory composite indicators' development? Is the selection of a logistics platform under sustainability perspectives affected by the compensation phenomenon?

To answer these research questions, this study aims to:

- Identify the sustainable evaluation criteria and sub-criteria for logistics platform location;
- Select the most suitable weighting and aggregation methods;
- Propose a composite indicator based on compensatory and partially compensatory multi-criteria decision support methods to identify the location of a logistics platform, responding more adequately to the requirements of sustainability;
- Study the impact of a compensation phenomenon on the decision-making process.

This manuscript is organized as follows. The next section reviews the relevant existing approaches for locating a logistics platform and developing composite indicators. Section 3 presents the proposed approach. Then, Section 4 illustrates the results of analyzing the case study applied in the city of Sfax. The implications of this study are presented in Section 5. Finally, the conclusion and future research directions are provided in Section 6.

## 2. Literature Review

The choice of a location is one of the problems discussed in the literature. We present, in the first subsection, an overview of the existing approaches to locating logistics platforms. The second subsection describes the steps of the construction of the composite indicators because it is useful tool that is increasingly demanded by policy-makers.

### 2.1. Existing Approaches of Locating Logistics Platform

Many methods, such as multi-criteria decision-making (MCDM) methods, meta-heuristics for multi-objective decision-making methods and multi-objective combinatorial optimization methods, have been applied to solve localization problems [7]. However, the proposed studies based on combinatorial optimization and meta-heuristics are more complex and do not always represent reality. In this regard, the authors of this study focused on multi-criteria decision support methods because they can provide insight about reality by integrating expert's opinions.

In fact, the facility location problem is one of the major problems discussed in the literature. This section describes the theoretical aspects of the suggested approach. It also presents some approaches proposed to solve the problem of localizing infrastructures using multi-criteria decision support methods. Table 1 describes the existing localization MCDM approaches.

Agrebi et al. [8] proposed a decision support system to select the location of distribution centers. The introduced multi-attribute and multi-actor decision-making method based on the Elimination and Choice Expressing Reality method (ELECTRE-I). Given the

inherent uncertainty and imprecision of human decision-making, a fuzzy multi-attribute and multi-actor decision-making method was also applied. To check the sensitivity of the chosen solution to variations in the criteria weights, a sensitivity analysis was carried out. The obtained results prove that the two suggested methods met the desired objective of the selection of the best location in a certain/uncertain context of multi-attribute and multi-actor variables.

Kumar et Anbanandam [9] presented a framework to select the location of the multi-modal freight terminal under sustainability perspectives. The authors used the intuitive fuzzy (IF) set to incorporate the importance of the expert's group decision-making process and calculated the priority weight of the criteria and its sub-criteria using the hierarchical analysis process (IF-AHP). Then, they evaluated the performance of location by incorporating the technique for order of preference by similarity to ideal solution (IF-TOPSIS).

Yazdani et al. [10] developed a two-step decision-making model to find the most preferred area for establishing logistics centers. In the first step, to identify the efficient and inefficient alternatives, the authors compared the considered communities through data envelopment analysis (DEA). However, in the second step, an evaluation model was designed to assess the performance of effective communities. Researchers employed the rough full consistency method (R-FUCOM), to obtain the optimal weights of the criteria, and the rough combined compromise solution (R-CoCoSo) method to rank efficient communities. The adopted model allowed for capturing the uncertainty and vagueness in the judgments of decision-makers by applying the rough set theory (RST). In addition, sensitivity analyses were performed to validate the robustness of the obtained results.

Cheng et Zhou [11] proposed a method to evaluate the location of the logistics distribution center. To improve the efficiency of decision-making, they developed a fuzzy approach based on the AHP method. Through a case study of four potential locations, the results prove that the adopted method is effective in selecting the best location with both qualitative and quantitative factors.

**Table 1.** The existing localization multi-criteria decision-making (MCDM) approaches.

| Authors | Weighting | Aggregation | Technical | Country | Extension |
|---------|-----------|-------------|-----------|---------|-----------|
| [9] | FI-AHP | FI-TOPSIS | Compensatory | India | Intuitionistic fuzzy sets |
| [10] | DEA, R-FUCOM | R-CoCoSo | Compensatory | - | Rough set theory |
| [8] | | ELECTRE I | Non-compensatory | - | Fuzzy sets |
| [12] | DEMATEL | MAIRCA | Compensatory | China | Fuzzy sets |
| [13] | DEMATEL | MAIRCA | Compensatory | China | - |
| [14] | GIS, Fuzzy SWARA | COCOSO | Compensatory | Turkey | Fuzzy sets |
| [15] | EW- Fuzzy AHP | Fuzzy TOPSIS | Compensatory | China | Fuzzy sets |
| [16] | Fuzzy AHP | PROMETHEE | Non-compensatory | Turkey | Fuzzy sets |
| [17] | DEMATEL, ANP | TOPSIS | Compensatory | Turkey | Intuitionistic fuzzy sets |

Pamucar et al. [12] applied a hybrid MCDM approach for the sustainable selection of a site to develop a multimodal logistics center. The decision-making trial and evaluation laboratory (DEMATEL) method was also used to determine the weight coefficients of the criteria. Then, a multi-attributive ideal-real comparative analysis (MAIRCA) was carried out to select a location by comparing the theoretical and empirical alternative assessments.

Muravev et al. [13] introduced a new approach based on DEMATEL-MAIRCA to determine the optimal locations of the China Railway Express international logistics centers and to minimize the number of rail routes. This approach found the best solution, closest to the ideal one. In the first instance, the value of the best alternative was identified in line with the observed criterion. In the second instance, the distances of the other alternatives were measured as a function of the observed criterion of the ideal value. The preliminary results showed that, because of the increase in the container turnover between China and the European Union, the determination of the optimal locations for the logistics centers should be done in a dynamic manner.

Ulutaş et al. [14] suggested a new approach combining the geographic information systems (GIS) with the fuzzy step-wise weight assessment ratio analysis (SWARA) method and the CoCoSo method to select the location of the logistics center in Turkey. In addition, sensitivity analyses were performed to validate the robustness of the suggested approach by varying criteria weights and comparing the ranking of the CoCoSo method with other MCDM techniques (complex proportional assessment of alternatives (COPRAS), VIseKriterijumska Optimizacija I Kompromisno Resenje (VIKOR), additive ratio assessment (ARAS), multi-objective optimization on the basis of ratio analysis (MOORA) and multi-attributive border approximation area comparison (MABAC)).

He et al. [15] developed a new hybrid fuzzy multi-criteria decision-making method to select the location of a joint distribution center by taking sustainability into account. First, the weights of the subjective criteria were calculated with a fuzzy AHP method, while the objective criteria were weighted using a fuzzy entropy method. Subsequently, the authors ranked the alternatives with the improved fuzzy TOPSIS method utilizing weighted criterion distances. Finally, a sensitivity analysis was carried out to illustrate the effectiveness and robustness of the proposed method and its ability to promote the sustainability of companies in China.

Kazançoğlu et al. [16] presented a hybrid multi-criteria decision-making approach to locate the logistics center in terms of benchmarks based on the sustainability concerns. This approach uses the fuzzy AHP method to obtain the weights of the criteria and the preference ranking organization method for enrichment evaluations (PROMETHEE) to select the best alternative. It was applied in Turkey, for the location of a sustainable logistics center in Izmir.

Karaşan et al. [17] adopted a new integrated fuzzy decision-making model to select the location of freight villages. In this model, the DEMATEL method was used to determine the most efficient criteria and their internal and external dependencies. The weight coefficients of the criteria were obtained using the analytic network process (ANP) method. Then, the TOPSIS method was employed to find the best alternative location. It was applied to a case study for the city of Istanbul in Turkey.

### 2.2. Composite Indicators

The literature of the existing works that proposed to solve the localization problem shows that, until now, no general or systematic method has been introduced to locate the logistics platform, and no methods have been specifically dedicated to the sustainability issue. Indeed, it is quite difficult to evaluate the choice of the platform location with several sustainable criteria. In this respect, the aggregation of these criteria in a composite indicator is an innovative and interesting approach that should be applied to localize logistics platforms. In this section, we present an overview of the composite indicators built, according to the literature, using MCDM methods.

The main procedures of building composite indicators include identifying sustainable criteria, weighting the identified criteria and aggregating these criteria into composite indicators [18–20]. Undoubtedly, the application of the weighting and aggregation methods is a key step in developing the composite indicators [18,19,21]. Controversial issues can arise at any stage of constructing the composite indicators. For this reason, the main challenge faced by the decision-maker is to choose the right weighting and aggregation methods that allow for the constructing of composite indicators [22,23].

In the literature, various weighting and aggregating MCDM methods were introduced to assess sustainability [20,23]. They are rather considered as a means of assisting decision-makers in developing composite indicators [23]. Weighting methods can be applied using several techniques such as AHP, ANP, the strengths, weaknesses, opportunities, threats (SWOT) method, SWARA, the best-worst method (BWM), the method of criterion impact loss (CILOS), the integrated determination of objective criteria weights (IDOCRIW), FUCOM, etc. As instances of aggregation methods, we can cite TOPSIS, VIKOR, DEMATEL, weighted aggregated sum product assessment (WASPAS), MAIRCA, CoCoSo,

PROMETHEE, ELECTRE, etc. Nonetheless, each method has specific characteristics and none of the MCDM methods can be applied to solve all types of problems of localizing infrastructures [24,25]. We present below an overview about the widely used weighting and aggregation methods. Then, we focus on three methods applied in the proposed approach.

### 2.2.1. Weighting Methods

One of the most important steps of constructing a composite indicator for the location of infrastructure is the weight of each criterion (called also indicator) [26]. More precisely, all the criteria may not have the same level of importance [27]. Thus, the weights can significantly affect the results of the overall composite indicators [28]. In fact, determining the weight of the criteria is one of the key problems that complicate the decision-making process. Weighting methods can be categorized into:

- Equal weighting methods [18];
- Objective data-based methods such as the principal component analysis (PCA) [29] and the data envelopment analysis (DEA) [18];
- Subjective participatory methods where the subjective opinions of experts and/or stakeholders are taken into account, such as the budget allocation (BAL) [19], AHP, FUCOM, etc.

Although there are several weighting methods, the AHP is the most intensively used together with recently-proposed methods such as BMW and FUCOM. Table 2 represents the most cited weighting methods.

**Table 2.** Analysis of the widely applied weighting methods.

| Methods | Characteristic | Simplicity | Comparison |
|---|---|---|---|
| AHP [30] | It defines the relationships between the different levels formed by a framework considered as an objective to be achieved. With AHP, it is almost impossible to make perfectly coherent pairwise comparisons with more than nine criteria. | Very critical | $n(n-1)/2$ |
| BWM [31] | It is based on a non-linear model used to determine the weights of the decision-making criteria by identifying the most preferable and least preferable criteria for making pairwise comparisons. | Average | $2n-3$ |
| FUCOM [32] | It allows for calculating weights and comparing criteria in pairs using integer, decimal or predefined scale values for the pairwise comparison of criteria. | Simple | $n-1$ |

### The FUCOM Method

The full consistency method (FUCOM) was recently developed by Pamucar et al. [32]. Thanks to its high stability, robustness and reliability, this method has quickly been applied in several works [33–43].

In a review done by Stojčić et al., the authors noticed that the AHP weighting method is the most implicated in solving problems related to transport and logistics [44]. However, Pamučar et al. [32] stated that the FUCOM is more consistent and preferable than the AHP method. The application of AHP method is somewhat complicated as it requires $n(n-1)/2$ pairwise comparisons of the criteria. On the other hand, FUCOM uses far fewer comparisons, which is one of its most important advantages. In addition, it provides the same results as those obtained by the BWM and AHP methods applying an integer or decimal scale. This method allows decision-makers to prioritize criteria utilizing a simple algorithm and applying an acceptable scale. Moreover, it makes it possible to obtain

the optimal weight coefficients with the possibility of validating them by showing the consistency of the results.

### 2.2.2. Aggregation Methods

Aggregation determines the mathematical operation of combining weighted criterion values [45]. In other words, in order to combine the different sustainability criteria, certain aggregation methods are needed [46]. When aggregating, data quality is of primary importance. Methods used to calculate the composite indicators can be classified into the following aggregation techniques [27,28,47]:

- Compensatory technique: It operationalizes the weak sustainability and allows for a high level of substitutability between criteria, which means that a poor performance in a criterion can be compensated by a good performance in another criterion. Otherwise, the weakness of one criterion could be hidden behind the strength of another criterion;
- Partially compensatory technique: This technique operationalizes the limited sustainability. It relies on geometric mean-based methods. In this case, a mutually preferential independence condition of indicators is required, with certain limits;
- Non-compensatory technique: It operationalizes the strong sustainability paradigm that partially or completely prevents the substitutability of criteria. Thus, an unfavorable result of one criterion cannot be compensated for by a favorable result of another criterion.

In the literature, several studies, such as those by the authors of [28,29,48] developed composite indicators with different levels of compensation. Moreover, other authors introduced non-compensatory composite indicators [49,50] and partial composite indicators [21,24,51]. Many studies [28,52] have picked up on this problem by discussing the compensatory issue. In this case, identifying dealt with this problem by discussing the compensatory issue. As such, identifying the appropriate aggregation technique to rank the alternatives is critical. The selection of aggregation techniques is one of the most contestable and scientifically relevant questions in the construction of the composite indicators [49]. After studying different MCDM methods used to aggregate the criteria, a comparative study of some aggregation methods was undertaken and the results are presented in Table 3. After that, the PROMETHEE and MAIRCA methods were chosen for various reasons:

- From a theoretical point of view, these methods use two different aggregation techniques. PROMETHEE and MAIRCA are based on compensatory aggregation and partially compensatory aggregation, respectively. Their objective is to study the impact of two sustainability perspectives: limited sustainability with the partially compensatory technique and weak sustainability with the compensatory technique utilized to choose a sustainable logistics platform;
- From a more practical point of view, these methods are known for their stability and robustness. The PROMETHEE and MAIRCA methods offer consistent solutions that do not change with the variation of the scale of values;
- Finally, because of their popularity [12,34,53–55], they were chosen to locate the logistics platform.

**Table 3.** Comparison between the MCDM methods.

| Methods | Type of Information | Stability | Simplicity | Technical |
|---------|--------------------|-----------|-----------| ----------|
| ELECTRE | Mixed | Medium | Moderately critical | Non-compensatory |
| PROMETHEE | Mixed | Stable | Moderately critical | Partially compensatory |
| MAIRCA | Mixed | Stable | Simple | Compensatory |
| TOPSIS | Quantitative | Medium | Moderately critical | Compensatory |
| VIKOR | Quantitative | Medium | Medium | Compensatory |

In the remainder of this section, we first define the two methods applied in this study. Then, we cite the advantages of each one while showing the reasons behind the choice

of these two methods. The objective of the joint use of both techniques is not to compare the two methods or to study the technical difference, but to help decision-makers deal with the main challenges of the sustainability perspectives when building the composite indicators. In other words, it aims to understand the location problem, which ultimately depends on the degree of inter-criteria compensation the decision-maker is willing to accept. The main challenge here is: how can the decision be made if different rankings were produced? In this case, it does not mean that one specific method is better than another. The decision-maker should deeply analyze the impact of sustainability on the location of the logistics platform. The objective of this study is to choose the location from the two-fold point of view of sustainability. The choice of one particular method depends on the type of solution expected by the decision-makers. More generally, the limited sustainability perspective is recommended when certain logistics platform localization projects have a good performance with limited compensatory aggregation that makes the rankings of alternatives robust. Consequently, F-PROMETHEE should be used. Furthermore, compensatory methods, such as F-MAIRCA, may be applied to select a second location if the decision-maker accepts changes in the sustainability strength.

Compensatory Aggregation Method: The MAIRCA Method

Multi Attributive Ideal-Real Comparative Analysis (MAIRCA) was developed in 2014 by the Center for Logistics Research of the University of Defense in Belgrade [56]. The MAIRCA has shown more stability, compared to other popular MCDM methods such as TOPSIS, ELECTRE, MOORA and COPRAS [12]. In fact, it is based on linear aggregation methods, which makes it a compensatory method. The MAIRCA was chosen because of its simple mathematical calculations, the stability of the solution and the possibility of combining this method with others [12]. The MAIRCA model is generally applied to assess the gap between the ideal and empirical assessments. The summation of the gaps in each criterion allows determining the total gap for each alternative. The ranking of the alternatives occurs at the end of the process where the highest ranked alternative is the one with the smallest gap value. The alternative with the lowest total gap value represents the alternative having values closer to the ideal scores (the ideal criteria values) [55].

Partially Compensatory Aggregation Method: The PROMETHEE Method

The PROMETHEE (preference ranking organization method for enrichment evaluations) method was developed to overcome the difficulties encountered in the implementation of the existing prioritization methods by Jean-Pierre Brans in the 1980s [57]. PROMETHEE has some limits in terms of compensation, particularly the absence of indifference and/or preference thresholds [28,54,58]. In this case, we used the usual function, characterized by a limited compensation degree and sustainability [53]. Thus, PROMETHEE is a partially compensatory technique. The PROMETHEE method is one of the most well-known and widely used upgrading methods. It can be extended to multi-decision-maker problems. This method is a potential tool to aid in-conflict resolution in a cooperative problem-solving environment. It is often chosen by decision-makers thanks to its easy implementation. It has the ability to imitate the human mind by making preferences among the alternatives in front of different contradictory criteria. Indeed, this method provides the decision-maker with a comprehensible representation and interpretation of the results.

## 3. Method

This article proposes composite indicators based on compensatory and partially compensatory multi-criteria decision support methods to identify the location of a logistics platform. The objective of formulating this composite indicator based on individual sustainability criteria and alternatives is to choose the best location by taking into account various sustainability considerations.

The human decision-making process is marked by imprecision and the inherent uncertainty associated with the lack of exact data of the criteria [59]. Indeed, with uncertainty, experts are generally unable to evaluate potential locations and to define the criteria weights, which underline the importance of fuzzy set theory in calculating the composite indicators [60]. Experts' uncertainty can also result when their preferences are often ambiguous and uncertain, which impacts the quality of the data resulting from their observations [8,61]. As decision-makers are unlikely to have full knowledge about all issues related to the choice of the location of a logistics platform, uncertainties need to be taken into account. For this reason, the current paper includes triangular fuzzy numbers given their ability to treat uncertainty about data in the decision-making process and reduce its complexity [61,62]. This form of fuzzy numbers is the most common one due to the simplicity and the rapidity of resolution [60,61].

The aim of this study is not to create a new fuzzy MCDM method, but to propose a methodological approach that allows decision-makers to face the main challenges of choosing the right MCDM method. Thus, it was necessary to study the impact of a compensation phenomenon to make solid decisions and ensure that the best localization alternative was selected. In order to solve this problem, a schematic representation of the suggested approach is presented in Figure 1. It consists of five main phases of calculating the composite indicator.

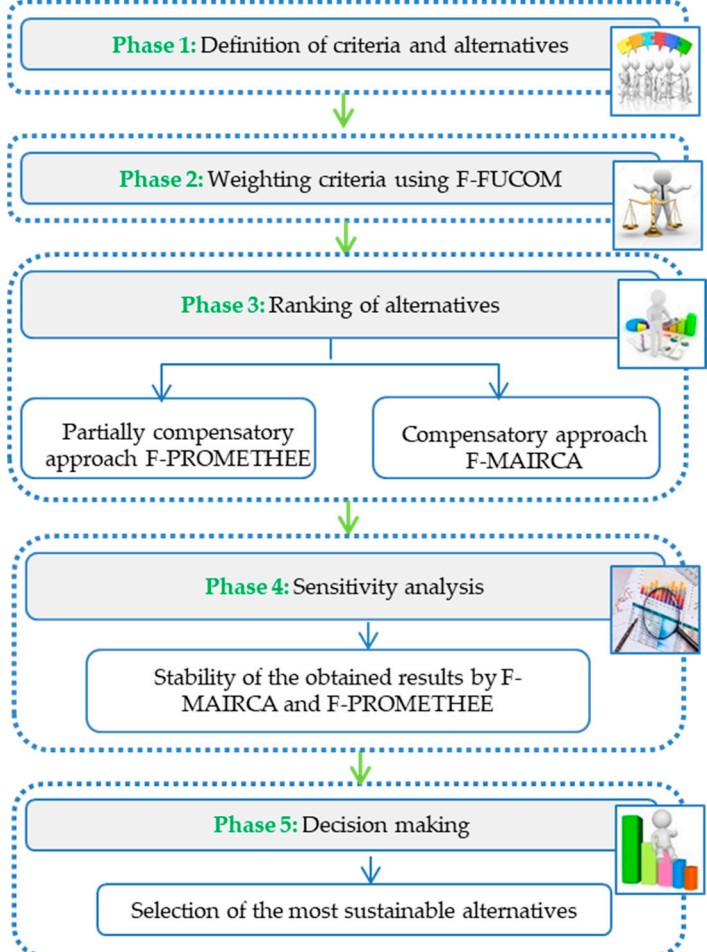

**Figure 1.** Schematic representation of the proposed approach.

### 3.1. Phase 1: Definition of Criteria and Alternatives

The first phase consists of identifying the criteria and alternatives. Indeed, the choice of location depends on several sustainability criteria such as social, territorial, economic

and environmental criterion, among others. To this regard, it seems important to select the most appropriate criteria.

In the literature, some studies [15,16,63] dealt with the problem of location from the sustainability point of view based on the three traditional criteria, namely, economic, environmental and social. In this context, this paper [15] integrates the economic, environmental and social criteria using 13 sub-criteria. The choice of these criteria was based on the literature and the experts' opinion about their project (City Joint Distribution for Online Shopping) in China. Kumar et Anbanandam [9] examined the choice of the multimodal freight terminal according to five criteria of sustainability: technical, economic, social, environmental and political. Thirteen sub-criteria implied by the authors of [11] were classified according to four criteria (economic, traffic, environmental and government policy). However, Muravev et al. [13] categorized the criteria used to locate the China Railway Express international logistics centers according to three types: social and economic criteria, geographical and infrastructure criteria and transport works criteria. Other researchers, such as the authors of [10,12,14,17], defined the criteria without classifying them according to sustainability criteria.

After analyzing the sustainability criteria discussed in the works presented above, the following section presents the criteria chosen to evaluate the location platform criteria, constructed under the traditional criteria of sustainability (economic, environmental and social) and emerging criteria (political and spatial). A fourth political criterion was introduced, given the interest of political criteria. This criterion represents the awareness and the required impact of local authorities on transport sustainability. A fifth territorial criterion was also suggested in order to choose the location of a logistics platform. Thus, the inclusion of this last criterion and its relationship with the traditional components of sustainability introduces the territorial cohesion perspective of sustainable development, i.e., a spatially equitable, efficient and consistent territory. Table 4 represents the different criteria assessed in this study.

### 3.1.1. Economic Criteria

Multimodal transport connectivity: Connectivity from the logistics platform to the city center is an important criterion [63]. A logistics platform should connect its location with other transport modes (motorways, rail, seaport, airport, etc.) to facilitate transit. Its proximity to other transport modes helps transport activities operate more efficiently [14].

Cost of land acquisition and construction: This cost must be properly controlled and minimized [64]. The selected location of the logistics platform varies depending on the land price [9], which is an important component of the total cost of the project [14,15]. Obviously, a higher land price will certainly increase the investment costs of building a logistic platform [63]. This cost is an essential component in the selection of the logistics platform [65].

Fiscal policy: Government fiscal support for the development of multimodal transport plays an essential role in stability in policy-making [9]. To attract investors and promote regional development, local and national authorities can offer fiscal advantages to investors [15]. Fiscal policy should be general for the city, imposing no spatial restrictions. In other words, the policy that favors a particular area of the city over other regions influences the chosen location. Fiscal policy should be general without the restriction of certain areas in the city that may influence the chosen location. Besides, governments should facilitate new platform developments via the implementation of new policies and by deregulation [66].

Transport cost: The terminal location must be close to the economic activities and industrial areas, which reduces the cost of short-distance transport and offers a competitive transport cost per ton-kilometer to ensure the easy flow of goods. The right choice of location may minimize the transport costs.

**Table 4.** The different criteria assessed in this study.

| Criteria | [9] | [10] | [11] | [12] | [13] | [14] | [15] | [16] | [17] | [63] | [62] | This Study |
|---|---|---|---|---|---|---|---|---|---|---|---|---|
| **Economic** | | | | | | | | | | | | |
| Multimodal transport connectivity | * | * | | * | | | | | | * | * | * |
| Cost of land acquisition and construction | * | * | * | | * | * | * | * | | * | * | * |
| Fiscal policy | * | | | | | | | | * | | | * |
| Transport costs | * | | * | | | | | | * | | | * |
| **Environmental** | | | | | | | | | | | | |
| Conformity with environmental emissions regulations | * | | | * | | | | | | * | | * |
| Effect on the natural landscape | * | | | | | | * | | * | * | | * |
| **Social** | | | | | | | | | | | | |
| Safety and security | | * | | * | * | | | | * | * | * | * |
| Noise | * | | | | | | | | * | | | * |
| Impact on nearby residents | | | | | | | * | | | * | | * |
| Impact on traffic congestion | * | | | | | | * | * | * | * | | * |
| **Political** | | | | | | | | | | | | |
| Current policy | * | | * | | | | | | | * | | * |
| Support role for industry | * | | | | | | * | * | | | | * |
| **Territorial** | | | | | | | | | | | | |
| Accessibility to multimodal transport | * | * | | * | | * | | * | | | * | * |
| Proximity to the industrial zone | * | * | | * | | * | | | * | | * | * |
| Possibility of extending the freight platform | * | * | | * | | | * | * | * | | * | * |

Abbreviation: * Sustainability criteria used in an existing study.

### 3.1.2. Environmental Criteria

Conformity with the environmental emissions regulations: The compliance of the logistics platform location with environmental laws and regulations is paramount. The protection of the natural environment and the reduction of urban pollution must also be considered [15]. The selected location should be in line with the spatial structure of the city and land-use planning [63]. Moreover, this platform should reduce the urban air pollution due to the movement of delivery vehicles. However, a bad choice of a location can create several problems of air pollution. This choice can generate transport flows and movement in the wrong place. Such criterion remarkably affects the environment because of its proximity to urban areas. Although the objective of setting up the platform is to encourage multimodality, efforts should nonetheless be directed toward making multimodal transport more sustainable by controlling air pollution and noise [66].

Effect on the natural landscape: The land chosen for the platform must promote harmony with the surrounding landscape. It must also maintain or enhance the original landscape without destroying vegetation and soil [15].

### 3.1.3. Social/Societal Criteria

Safety and security: the logistics platform has to protect persons against accidents, theft and vandalism [62,63], hence, the importance of the political reform efforts to promote platform security [9].

Noise: The noise generated by the movement of vehicles has a negative impact on the environment. A proposed location should use low-noise equipment to mitigate the noise impact of freight movement on the surrounding area [9]. The noise in a platform and in its surroundings varies from one location to another, where special attention should be paid to reduce noise pollution.

Impact on nearby residents: The choice of location must integrate the social environment. It should not only reduce disruption to urban life, but also relieve pressure on urban congestion and promote healthy development [63]. A location should also be far from densely-populated places to prevent accident occurrence [14].

Impact on traffic congestion: It is important to anticipate the influence of the selected infrastructure on traffic. For this reason, choosing the wrong location of the logistics platform can deteriorate local traffic conditions. To ensure the proper functioning of the logistics platforms, the surrounding traffic environment must be organized [63].

### 3.1.4. Political Criteria

Current policy: Current policy is included as it is an essential requisite to locate a logistics platform for a solid and sustainable base. A stable government system develops coherent policies for the development of a multimodal system in the whole nation. In fact, political stability plays a crucial role in providing the stability of policy-making [9]. The impact of localization on the city should be examined as part of the current transport policy. In the case where current policy criterion give particular importance to the specific location, regardless of the whole city, this criterion depends on the location compared to another one. Moreover, strong and stable policy is one of the important factors that needs to be considered in locating a logistics platform [66].

Support role for industry: The local government should establish appropriate policies to promote the development of its industry in logistics centers by taking sustainability into account [8,14,15]. Indeed, giving more support to one location by authorities, without considering the stakeholders' views, can influence the choice of location. A fundamental part of sustainable decision-making at the policy level concerns the degree of participation in the locating of a logistics platform [67]. More precisely, the support and the inclusion of relevant stakeholder groups (such as shippers, logistics service providers, receivers, etc.) is important for choosing the best location. Both coordination and cooperation between the government and industry are indispensable to avoid conflicts of interest in selecting the appropriate location [15,66].

### 3.1.5. Territorial Criteria

Accessibility to multimodal transport: Reduced accessibility weighs on the development of economic centers. Therefore, the location of the logistics platform should be connected to all modes of transport [15,16,63,68].

Proximity to the industrial zone: Proximity to the freight market, to the production area and to freight shippers is considered as the most critical criterion that, in turn, reduces the costs of transportation. A platform should be at the service of companies operating in different sectors. Therefore, an infrastructure location should be close to the industrial area [14,68].

Possibility of extending the freight platform: The availability of suitable land for the development of infrastructure is essential. It requires more land to ensure the possibility of extending the freight platform. The platform size should increase by building a new container yard, warehouses, a parking lot, etc. [62,68].

The sustainability criteria chosen to evaluate the location of the logistics platform are shown in Table 5. They can be classified into cost (C) or benefit (B) criteria. More precisely,

cost criteria should be minimized, i.e., the lower the value of the criterion is, the better the alternative will be. However, profit criteria must be maximized, that is to say, the higher the criterion value is, the better the alternative will be.

**Table 5.** Evaluation criteria and sub-criteria.

| Unit | Criteria | Definition | Type |
|---|---|---|---|
| $C_1$ | Economic | | |
| $C_{1.1}$ | Multimodal transport connectivity | Connectivity of the location to other modes of transport, e.g., highways, railways, seaport, airport, etc. | Benefit |
| $C_{1.2}$ | Cost of land acquisition and construction | The location of a logistics platform to be selected depends on these costs, which must be properly controlled and minimized. | Cost |
| $C_{1.3}$ | Fiscal policy | The fiscal advantages offered by the authorities to attract investors and promote the development of transport. | Cost |
| $C_{1.4}$ | Transport cost | The location should be close to the source of freight to reduce the cost of transportation. | Cost |
| $C_2$ | Environmental | | |
| $C_{2.1}$ | Conformity with environmental emissions regulations | Choosing the right location can reduce the impact of air pollution on human health and the environment. | Benefit |
| $C_{2.2}$ | Effect on the natural landscape | To promote harmony with the surrounding landscape without destroying the original landscape. | Cost |
| $C_3$ | Social | | |
| $C_{3.1}$ | Safety and security | The platform is protected against accidents, theft and vandalism. | Benefit |
| $C_{3.2}$ | Noise | The noise generated by the movement of vehicles has a negative impact on environments. | Cost |
| $C_{3.3}$ | Impact on nearby residents | A location should promote healthy development for urban residents. | Cost |
| $C_{3.4}$ | Impact on traffic congestion | Traffic environment planning to relieve pressure on urban congestion. | Cost |
| $C_4$ | Political | | |
| $C_{4.1}$ | Current policy | Political stability plays a crucial role in the stability of the development of a multimodal system. | Benefit |
| $C_{4.2}$ | Support role for industry | The local government should establish appropriate policies to promote the development of its industry in platforms. | Benefit |
| $C_5$ | Territorial | | |
| $C_{5.1}$ | Accessibility to multimodal transport | A location should be connected and accessible to all modes of transport. | Benefit |
| $C_{5.2}$ | Proximity to the industrial zone | A platform should be at the service of companies operating at different sectors. | Benefit |
| $C_{5.3}$ | Possibility of extending the freight platform | The infrastructure must have the capacity to increase the size of the platform, to meet the growing freight demands. | Benefit |

### 3.2. Phase 2: Weighting of Criteria Using FUCOM

The second phase was necessary to calculate the weight of the criteria and sub-criteria using the fuzzy FUCOM method (F-FUCOM). In the following section, we present the F-FUCOM algorithm whose application consists in the following five steps.

**Step 1:** Ranking the criteria according to their level of importance. In the first step, the ranking was evaluated according to the importance of the evaluation criteria $C = \{C_1, C_2, \ldots, C_n\}$. This means the first rank was assigned to the criterion that was expected to have the highest weight coefficient. The last place was occupied by the criterion

for which we expected the lowest value of the weight coefficient. The criteria classified according to the expected values of weight coefficients are presented as follows:

$$C_{j(1)} \succ C_{j(2)} \succ \ldots \succ C_{j(n)} \tag{1}$$

where $C_{j(1)}$ and $C_{j(n)}$ respectively stand for the most and the least important criterion among the predefined set of n elements. Notably, if two or more criteria have equal importance, the equality sign "=" is placed between the criteria instead of ">".

**Step 2:** Determining the comparative priority of the criteria. The mutual comparison of the criteria was carried out using fuzzy linguistic expressions from the defined scale. Mutual comparisons were made by each expert individually {E1, E2, ... , Et} according to his/her preferences. The comparison was performed using the first-ranked (most important) criterion. Thus, for each expert, the fuzzy criteria significance was obtained for all the criteria ranked in Step 1. Based on the defined significances of the criteria and using Equation (2), the fuzzy comparative significance $\varphi_{j/(j+1)}$ was determined.

$$\varphi_{j/(j+1)} = \frac{\varpi_{Cj}}{\varpi_{C(j+1)}} = \left( \frac{w_j^l}{w_{j+1}^u}, \frac{w_j^m}{w_{j+1}^m}, \frac{w_j^u}{w_{j+1}^l} \right) \tag{2}$$

The fuzzy vectors of the comparative significance of the criteria for each individual expert {E1, E2, ... , Et} were obtained applying Equation (3), where e = (1, 2, ..., t).

$$\Phi^e = \left( \varphi_{1/2}, \varphi_{2/3}, \ldots, \varphi_{(n-1)/n} \right) \tag{3}$$

**Step 3:** Defining the limits of the fuzzy model. The final values of weight coefficients should satisfy two constraints:

- **Constraint 1**: the ratio of the weights of the criteria should be the same as their comparative signification between the observed criteria.

$$\varphi_{j/(j+1)} = \frac{w_j}{w_{j+1}} \tag{4}$$

- **Constraint 2**: The final values of weight coefficients should satisfy the transitivity condition, respectively $\varphi_{\frac{j}{j+1}} \otimes \varphi_{\frac{j+1}{j+2}} = \varphi_{\frac{j}{j+2}}$, i.e., $\frac{w_j}{w_{j+1}} \otimes \frac{w_{j+1}}{w_{j+2}} = \frac{w_j}{w_{j+2}}$. This second condition must fulfill the final values of weight coefficients.

$$\frac{w_j}{w_{j+2}} = \varphi_{j/(j+1)} \otimes \varphi_{(j+1)/(j+2)} \tag{5}$$

**Step 4:** Designing a fuzzy model to calculate the optimal values of the weights of the criteria. In this step, the final values of the fuzzy weight coefficients of the criteria were computed for each expert: $(w_1, w_2, \ldots, w_n)^e$. The conditions defined in the third step should be met with the minimal deviation from the maximal consistency. In other words, the conditions must be met $\frac{w_j}{w_{j+1}} - \varphi_{j/(j+1)} = 0$ and $\frac{w_j}{w_{j+2}} - \varphi_{j/(j+1)} \otimes \varphi_{(j+1)/(j+2)} = 0$. Under these conditions, the maximum coherence amounts to $\chi = 0$.

In order to satisfy the previously presented conditions, the non-linear model was applied to determine the optimal fuzzy values of the weight coefficients for the evaluation criteria, which can be formulated as follows:

$$
\min \chi
$$
$$
\text{s.t.}
$$
$$
\begin{cases}
\left| \frac{w_j}{w_{j+1}} - \varphi_{j/(j+1)} \right| \leq \chi \,, \ \forall j \\
\left| \frac{w_j}{w_{j+2}} - \varphi_{j/(j+1)} \otimes \varphi_{(j+1)/(j+2)} \right| \leq \chi \,, \ \forall j \\
\sum_{j=1}^{n} w_j = 1, \ \forall j \\
w_j^l \leq w_j^m \leq w_j^u \\
w_j^l \geq 0 \,, \ \forall j \\
j = 1, 2, \ldots, n
\end{cases}
\tag{6}
$$

where $w_j = \left( w_j^l, w_j^m, w_j^u \right)$ and $\varphi_{j/(j+1)} = \left( \varphi_{j/(j+1)}^l, \varphi_{j/(j+1)}^m, \varphi_{j/(j+1)}^u \right)$

Considering that the maximum consistency requires fulfilling the condition where $\frac{w_j}{w_{j+1}} - \varphi_{j/(j+1)} = 0$ and $\frac{w_j}{w_{j+2}} - \varphi_{j/(j+1)} \otimes \varphi_{(j+1)/(j+2)} = 0$, model (6) can be transformed into a fuzzy linear model (7) and, by solving it, the optimal fuzzy values of the weight coefficients were obtained.

$$
\min \chi
$$
$$
\text{s.t.}
$$
$$
\begin{cases}
\left| w_j - w_{j+1} \otimes -\varphi_{j/(j+1)} \right| \leq \chi \,, \ \forall j \\
\left| w_j - w_{j+2} \otimes \varphi_{j/(j+1)} \otimes \varphi_{(j+1)/(j+2)} \right| \leq \chi \,, \ \forall j \\
\sum_{j=1}^{n} w_j = 1 \,, \ \forall j \\
w_j^l \leq w_j^m \leq w_j^u \\
w_j^l \geq 0 \,, \ \forall j \\
j = 1, 2, \ldots, n
\end{cases}
\tag{7}
$$

where $w_j = \left( w_j^l, w_j^m, w_j^u \right)$ and $\varphi_{j/(j+1)} = \left( \varphi_{j/(j+1)}^l, \varphi_{j/(j+1)}^m, \varphi_{j/(j+1)}^u \right)$

**Step 5**: Calculating the final optimal values of the criteria weights. Solving model (6) or (7) allowed us to obtain the weight coefficients of criteria by the experts.

### 3.3. Phase 3: Ranking of Alternatives

The third phase aims at ranking the different alternatives. We used fuzzy MAIRCA (F-MAIRCA) and fuzzy PROMETHEE (F-PROMETHEE) to evaluate the performance of the location considering a compensatory and partially compensatory reasoning, respectively. The PROMETHEE method is based on the partially compensatory aggregation technique (an unfavorable result of one criterion can be limited, compensated by a favorable result of another). However, the MAIRCA method relies on the compensatory aggregation technique (the weakness of one criterion could be hidden behind the strength of another).

### 3.3.1. Compensatory Approach: The F-MAIRCA Method

**Step 1:** Building the initial decision matrix, $\widetilde{D}^t$. On the basis of the linguistic evaluation of alternatives with respect to the considered criteria, the matrix $\widetilde{D}^t = \left[\widetilde{A}_{ij}^{(t)}\right] \forall i = 1, 2, \ldots, m$ and $j = 1, 2, \ldots, n$ was constructed as shown in Equation (8).

$$\widetilde{D}^t = \begin{bmatrix} \widetilde{A}_{11}^{(t)} & \cdots & \widetilde{A}_{1n}^{(t)} \\ \vdots & \ddots & \vdots \\ \widetilde{A}_{m1}^{(t)} & \cdots & \widetilde{A}_{mn}^{(t)} \end{bmatrix} \tag{8}$$

where $\widetilde{A}_{mn}^{(t)}$ demonstrates that the m-th alternative was evaluated linguistically with respect to the n-th criterion by the t-th expert.

**Step 2:** Building the fuzzy aggregate decision matrix. The fuzzy aggregated decision matrix was constructed using the arithmetic operator, as represented in Equation (9).

$$\widetilde{D} = \begin{bmatrix} \widetilde{A}_{11} & \cdots & \widetilde{A}_{1n} \\ \vdots & \ddots & \vdots \\ \widetilde{A}_{m1} & \cdots & \widetilde{A}_{mn} \end{bmatrix} \text{ where } \widetilde{A}_{11} = \frac{\widetilde{A}_{11}^{(1)} + \widetilde{A}_{11}^{(2)} + \cdots + \widetilde{A}_{11}^{(t)}}{t} \tag{9}$$

**Step 3:** Determining the preferences of the alternatives. In the third step, we defined preferences in relation to the selection of alternatives. The decision-maker was neutral, i.e., he/she had no preference towards any of the suggested alternatives. Since there was an equal probability between the alternatives, the preferences for each of them can be defined as follows:

$$P_{Ai} = \frac{1}{m};$$
$$\sum_{i=1}^{m} P_{Ai} = 1; \ i = 1, 2, \ldots, m \tag{10}$$

where m is the number of alternatives.

**Step 4:** Determining the fuzzy matrix of the theoretical ponder, $\widetilde{T}_{PA}$. This matrix was obtained by multiplying the preferences of alternatives and fuzzy criteria weights provided by FUCOM. The fuzzy matrix of theoretical ponder was calculated as demonstrated in Equation (11).

$$\widetilde{T}_{PA} = \begin{bmatrix} P_{Ai}\widetilde{w}_1 & \cdots & P_{Ai}\widetilde{w}_n \\ \vdots & \ddots & \vdots \\ P_{Ai}\widetilde{w}_1 & \cdots & P_{Ai}\widetilde{w}_n \end{bmatrix} = \begin{bmatrix} \widetilde{t}_{P_{11}} & \cdots & \widetilde{t}_{P_{1n}} \\ \vdots & \ddots & \vdots \\ \widetilde{t}_{P_{m1}} & \cdots & \widetilde{t}_{P_{mn}} \end{bmatrix} \tag{11}$$

**Step 5:** Constructing the normalized fuzzy aggregated decision matrix. This step consisted of normalizing the fuzzy aggregated decision matrix, as defined in Step 2. This matrix can be expressed as follows: $X = \left[X_{ij}\right]_{m,n} \forall i = 1, 2, \ldots, m$ and $j = 1, 2, \ldots, n$ [69]. If C and B represent the cost and benefit criteria, respectively, then the normalization procedure is as presented in Equations (12) and (13).

$$X_{ij} = \left(\frac{a_j^{L-}}{a_{ij}^N}, \frac{a_j^{L-}}{a_{ij}^M}, \frac{a_j^{L-}}{a_{ij}^L}\right), \ j \in C, \ a_j^{L-} = \min_i a_{ij}^L \tag{12}$$

$$X_{ij} = \left(\frac{a_{ij}^L}{a_j^{N+}}, \frac{a_{ij}^M}{a_j^{N+}}, \frac{a_{ij}^N}{a_j^{N+}}\right), \ j \in B, \ a_j^{N+} = \max_i a_{ij}^N \tag{13}$$

**Step 6:** Calculating the matrix of fuzzy actual ponder , $\widetilde{T}_{RA}$. This matrix was obtained by multiplying the fuzzy normalized decision matrix, as shown in Step 5, and the fuzzy of theoretical ponders, as revealed in Step 4.

$$\widetilde{T}_{R_A} = \begin{bmatrix} \widetilde{t}_{R_{11}} & \cdots & \widetilde{t}_{R_{1n}} \\ \vdots & \ddots & \vdots \\ \widetilde{t}_{R_{m1}} & \cdots & \widetilde{t}_{R_{mn}} \end{bmatrix} = \begin{bmatrix} \widetilde{n}_{11} \times \widetilde{t}_{P_{11}} & \cdots & \widetilde{n}_{1n} \times \widetilde{t}_{P_{1n}} \\ \vdots & \ddots & \vdots \\ \widetilde{n}_{m1} \times \widetilde{t}_{P_{m1}} & \cdots & \widetilde{n}_{mn} \times \widetilde{t}_{P_{mn}} \end{bmatrix} \tag{14}$$

**Step 7:** Calculating the total gap matrix. In this step, the Euclidian distance between the matrix of fuzzy theoretical ponder and the actual ponder of each alternative with respect to each criterion was calculated. In other works [13,70,71], the authors suggested subtracting $T_{PA}$ and obtaining the total gap matrix $G$. It is preferable that the highest rank alternative should have a minimum gap value from each criterion. The calculation was done as shown in Equation (15).

$$g_{ij} = \sqrt{\frac{1}{3}\left[\left(t^l_{P_{ij}} - t^l_{R_{ij}}\right)^2 + \left(t^m_{P_{ij}} - t^m_{R_{ij}}\right)^2 + \left(t^u_{P_{ij}} - t^u_{R_{ij}}\right)^2\right]} \tag{15}$$

**Step 8:** Summating the gap values and ranking the alternatives. The gap values $g_{ij}$ were summed for each alternative with respect to each criterion by using Equation (16). The final values were then arranged in ascending order, and finally, the preferences were ranked.

$$Q_i = \sum_{j=1}^{n} g_{ij}, \quad i = 1, 2, \ldots, m \tag{16}$$

3.3.2. Partially Compensatory Approach: The F-PROMETHEE Method

**Step 1:** Creating a decision matrix. Each decision-maker evaluated n criteria according to m alternatives (or actions). Where j = 1,2, … ,n and i = 1,2, … ,m.

**Step 2**: Constructing the standard fuzzy performance matrix. The normalization of the comparison values of the decision matrix was obtained by deploying the expressions given in Equations (12) and (13) to construct a normalized fuzzy performance matrix.

**Step 3:** Determining the preferred function. The preference function between the two alternatives (a) and (b) was defined by the indifference and preference thresholds. The latter, denoted (p), is the lowest value of $d_j(a, b)$, below which there is indifference between selecting (a or b). The indifference threshold (q) is the lowest of $d_j(a, b)$, and there is strict preference of (a over b). Six different types of preference functions were proposed, namely, Usual (Type I), quasi-criterion (Type II), linear preference criterion (Type III), level criterion (Type IV), linear preference criterion and zone of indifference (Type V), as well as the Gaussian criterion (Type VI) [57]. A preference function was defined for each criterion using Equation (17), where (a) and (b) are two alternatives of the set of (m) alternatives.

$$d_j(a, b) = l_{aj} - u_{bj}; m_{aj} - m_{bj}; u_{aj} - l_{bj} \tag{17}$$

It is preferable that experts define the preference function. However, this step makes the application of the method more cumbersome and complex. To deal with this limitation, we used the first assumption of the strict preference function because it is the most widely used in the literature [69,72,73]. The choice of the usual preference function is based on the need to use an assessment model that can be easily understood by decision-makers. On the other hand, this function does not include any threshold values, which is typically a complex and time-consuming exercise for decision-makers [53]. Regardless of the difference between the alternatives, with this strict preference function, the best evaluated alternative is always the most preferred. However, the effect of the types of preference functions of PROMETHEE on the final preferences is not known. The choice of the usual function could

be a limitation of this study. The usual function is without an indifference threshold. With this function, $F_j[d_j(a, b)] = 0$ when $d_j(a, b) \leq 0$ and $F_j[d_j(a, b)] = 1$ when $d_j(a, b) > 0$.

**Step 4:** Calculating the preference index for each criterion. The preference index, $P_j(a, b)$, which denotes the preference of (a) over (b) for criterion j, was calculated as a function of $d_j(a, b)$ by applying Equation (18). The value of $P_j(a, b)$ varies from 0 to 1 such that [47]:

- $P(a, b) = 0$ means an indifference between (a) and (b) or no preference of (a) over (b);
- $P(a, b) \sim 0$ means a weak preference of (a) over (b);
- $P(a, b) \sim 1$ means a strong preference of (a) over (b);
- $P(a, b) = 1$ means a strict preference of (a) over (b).

$$P_j(a, b) = \left( F_j(l_{aj} - u_{bj}); F_j(m_{aj} - m_{bj}); F_j\left(u_{aj} - l_{bj}\right) \right) P_j(a, b) = (l^j_{ab}, m^j_{ab}, u^j_{ab}) \quad (18)$$

**Step 5:** Calculating the overall preference index. A multi-criteria degree of preference was then calculated to globally compare each pair of shares. The global preference index represents the degree of preference of alternatives (a) over (b), considering all the criteria simultaneously [57]. The fuzzy global preference index $\pi$ (a,b) was calculated by applying Equation (19) with the assumption that the relative weights of the criteria were also triangular fuzzy numbers, $W_j = (l''_j, m''_j, u''_j)$.

$$\pi(a, b) = \sum_{j=1}^{n} w_j \otimes P_j(a, b) \, \pi(a, b) = \sum \left( l''_j, m''_j, u''_j \right) \otimes (l^j_{ab}, m^j_{ab}, u^j_{ab}) \, \pi(a, b) = (l^\pi_{ab}, m^\pi_{ab}, u^\pi_{ab}) \quad (19)$$

where $\pi(a, b) \approx 0$ denotes a low preference of (a) over (b) and $\pi(a, b) \approx 1$ denotes a strong preference of (a) over (b).

**Step 6:** Determining the positive outranking flow $(\Phi+)$ and negative outranking flow $(\Phi-)$. The outgoing and incoming flows for each alternative were calculated using the respective Equations (20) and (21) [74]. The leaving flow represents the dominance of one action (a) over the other actions. Thus, the entering flow represents the weakness of an action (a) compared to other actions.

$$\Phi^+(a) = \frac{1}{n-1} \sum \pi(a, x) , \, x \in (i = 1 \ldots m) \quad (20)$$

$$\Phi^-(a) = \frac{1}{n-1} \sum \pi(x, a), \, x \in (i = 1 \ldots m) \quad (21)$$

**Step 7:** Determining the partially priorities using PROMETHEE I. The partial priorities include three possible outcomes: the preference of one decision point over another, indifference between the decision points and inability to compare the decision points with each other.

$$
\begin{cases}
aPb & \begin{cases} \Phi^+(a) > \Phi^+(b) \text{and} \Phi^-(a) < \Phi^-(b) \text{or} \\ \Phi^+(a) = \Phi^+(b) \text{and} \Phi^-(a) < \Phi^-(b) \text{or} \\ \Phi^+(a) > \Phi^+(B) \text{and} \Phi^-(a) = \Phi^-(b) \end{cases} \\
aIb & \{ \Phi^+(a) = \Phi^+(b) \text{and} \Phi^-(a) = \Phi^-(b) \\
aRb & \begin{cases} \Phi^+(a) > \Phi^+(b) \text{and} \Phi^-(a) > \Phi^-(b) \text{or} \\ \Phi^+(a) < \Phi^+(b) \text{and} \Phi^-(a) < \Phi^-(b) \end{cases}
\end{cases} \quad (22)
$$

where (P, I, R) respectively denote preference, indifference and incomparability in PROMETHEE.

**Step 8:** Determining all priorities applying PROMETHEE II. The PROMETHEE II method provides a total pre-order (excludes incomparability and considerably reduces indifference). The net flow is the subtraction of $\Phi^-(a)$ and $\Phi^+(a)$ (Equation (23)). At this step, all the alternatives become comparable, as no incomparability remains. They could thus be classified by full ranking them. The best alternative is the one with the highest net

flow. At the end of the calculations, the fuzzy net flow values obtained could be defuzzified to facilitate the comparisons of alternatives.

$$\Phi(a) = \Phi^+(a) - \Phi^-(a) \begin{cases} aPb \text{ if } \Phi(a) > \Phi(b) \\ aIb \text{ if } \Phi(a) = \Phi(b) \end{cases} \tag{23}$$

### *3.4. Phase 4: Sensitivity Analysis*

In the fourth phase, a sensitivity analysis was performed to rigorously test the robustness and the feasibility of the proposed approach. This analysis allows the decision-maker to check whether the final ranking is sensitive, and depends on the approach adopted to aggregate the scores of the criteria.

#### 3.4.1. Assessment of the Independence of the Aggregation Technique

The objective of this analysis is to evaluate the stability of a solution by changing the applied aggregation technique. Three different methods (F-PROMETHEE, F-MAIRCA and F-TOPSIS) based on compensation techniques were used. Moreover, three sustainability perspectives were considered: limited sustainability with the use of the partial compensation technique, weak sustainability with the use of the compensation technique and strong sustainability with the use of the non-compensation technique. The authors of [21] mentioned that the use of the three aggregation techniques is strongly recommended to adequately assess the obtained results. The type of sustainability is directly related to the aggregation approach and the level of compensability. For this reason, F-TOPSIS was applied using the non-compensation technique (an unfavorable result of one criterion cannot be compensated by a favorable result of another) (Garcia-Bernabeu et al.'s method [21]).

#### 3.4.2. Variation of Criteria Weights

Sensitivity analysis is generally performed to assess the influence of the weights assigned by experts on the ranking of alternatives. The experts' preferences are different. In fact, some give more importance to one criterion over another. In addition, the same criterion may be given different weights by the same expert in different situations. Therefore, the authors of [70] suggested checking the final ranking of the alternatives with a small variation of criteria weights. The sensitivity analysis was carried out in the experiments through 90 scenarios divided into three phases. In phase 1, the weight coefficients of the criteria in 30 scenarios increased or decreased by 45%. In each of the 30 scenarios, one weight coefficient increased by 45%, and it was favored. In the same scenario, the weight coefficients of the remaining criteria decreased by 45%. In phase 2 a similar procedure was applied in the next 30 scenarios, with weight coefficients that increased or decreased by 65%. Finally, in phase 3, the weight coefficients increased or decreased by 85%.

### *3.5. Phase 5: Decision-Making Process*

In the last phase, the potential alternatives were ranked to select the most sustainable location of the logistics platform. In this phase, decision-makers can also identify the impact of criteria on the selection of the platform.

## 4. Results

### *4.1. Problem Definition and Alternatives Selection*

In the following section, the proposed approach, applied to locate the logistics platform, is described taking sustainability into account. This study was carried out in the city of Sfax. In Tunisia, road transport is the main means of the routing of goods. According to the Ministry of Transport, it concentrates 85% of overland flows of goods. More precisely, freight transport is at the heart of economic and social development. Despite its preponderant place, freight transport generates various problems, especially in cities where the flow of distribution is important. In this study, we chose the city of Sfax because it is known by its significant economic dynamism and rich industrial fabric. In fact, it is

the second largest Tunisian city after the capital Tunis, in terms of both its demographic weight and industrial activities. It is located in the center-east of the Republic of Tunisia. Its privileged geographical position, its wide opening to the sea and its port make it a national and international commercial and trading center. In other words, it constitutes a natural corridor for the transport of goods. However, it includes a set of elements and issues that directly or indirectly influence the efficiency of the transport system. The city of Sfax, despite its economic and industrial dynamism, has suffered for decades from serious problems in the transport and circulation system, which affect sustainable development. During peak hours (morning, noon and evening) on the eve of the holidays, the difficulty of transport becomes even more difficult. These congestion and traffic problems are due to the spatial concentration of economic activities and administrative services. They are aggravated by the increase of heavy goods vehicles traffic and the architecture of the road infrastructure. Moreover, there is a strong embarrassment occasioned by flows freight transport that have an origin or destination at the commercial port or in the industrial zones, which are predominantly located in the coastal wings.

For years, this city has suffered greatly from this system, which is unsuited to the country's economic, demographic and urban growth. Further, transport and traffic conditions in Sfax have deteriorated significantly in recent years. Thus, with the intensification of the national and international containerized exchanges, the implementation of a multimodal infrastructure can contribute to the sustainable development of the city and remedy the problems mentioned below. As a solution, a better location of the logistics platform would ensure the interconnection of the available transport infrastructure to improve its performance. Moreover, this infrastructure plays a fundamental role in the development of the regional and extra-regional economic activities by affecting both the prices of production and the sale of goods.

The location of the logistics platform in the city of Sfax has been given great interest for several years. However, the identification of locations depends on the preferences of the decision-makers and the conditions of freight transport. After interviewing several decision-makers, seven potential locations were considered. These locations were defined as alternatives that meet the interests of all the stakeholders in the city. The different alternatives, chosen in Sfax, are as follows (Figure 2): (A1) GARGROUR, (A2) NAKATA, (A3): ELGONNA, (A4) ELHAJEB1, (A5) ELHAJEB2, (A6) LA SIAPE and (A7) SKHIRA.

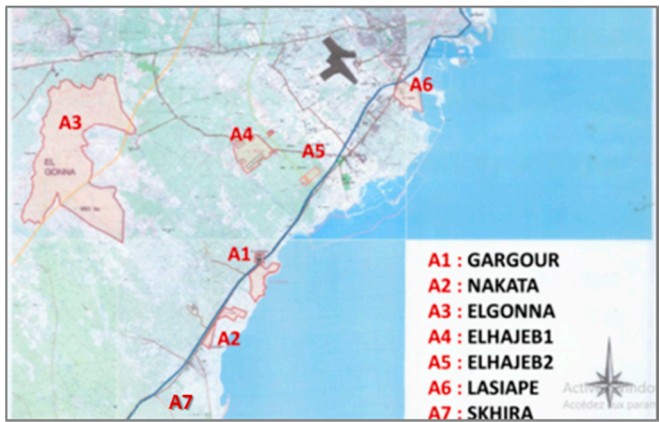

**Figure 2.** Set of alternatives for logistics platform location in Sfax.

### 4.2. Weighting of Criteria

#### 4.2.1. Obtaining Linguistic Judgments

In this step, we interviewed seven experts {E1, E2 . . . E7} involved in the decision-making process. These experts all had at least 10 years of experience. Table 6 shows the characteristics of the consulted experts. They participated in the process of determining the weights for the criteria and evaluating the alternatives. A given 5-point scale was used

in the survey to establish the experts' fuzzy language ratings. Data were collected in the city of Sfax between July and August 2020.

**Table 6.** Characteristics of the experts.

|  | Age | Level of Education | Experience | Profession | Organization Name |
|---|---|---|---|---|---|
| E1 | 55 years | PhD degree in electrical engineering | 25 years | Regional Director of Transport in Sfax | Ministry of transport |
| E2 | 35 years | Industrial engineer | 10 years | Logistics Director | SOCOMENIN |
| E3 | 49 years | Civil engineer | 23 years | Port Director | Office merchant marine and Ports of Sfax |
| E4 | 43 years | Master's degree in international trade | 18 years | Logistics Director, contractual teacher | Pastry MASMOUDI |
| E5 | 58 years | PhD in urban planning | 28 years | Municipal civil servant | Municipality of Sfax |
| E6 | 49 years | Master in business strategy | 23 years | Port Technical Director, temporary teacher | Office merchant marine and Ports of Sfax |
| E7 | 36 years | Master's degree in logistics | 11 years | Administrator | Governorate of Sfax |

### 4.2.2. The F-FUCOM Results

The C1–C5 criteria constitute the first hierarchical level, while the second hierarchical level consists of the sub-criteria, classified into the five criteria presented in Table 5. Using fuzzy FUCOM, the values of the local weights of the sub-criteria were calculated. After defining the local weights of the sub-criteria, the weights of the criteria were multiplied by the group of weights of the sub-criteria. We obtained the global values, used subsequently in this analysis, to evaluate alternatives with MAIRCA and PROMETHEE. In order to solve this problem, six fuzzy FUCOM models were defined:

- Model 1: Calculation of the values of the weight coefficients of the criteria C1, C2, C3, C4 and C5;
- Model 2: Calculation of the local values of the weight coefficients of the sub-criteria C1.1, C1.2, C1.3 and C1.4;
- Model 3: Calculation of the local values of the weight coefficients of the sub-criteria C2.1 and C2.2;
- Model 4: Calculation of the local values of the weight coefficients of the sub-criteria C3.1, C3.2; C3.3 and C3.4;
- Model 5: Calculation of the local values of the weight coefficients of the sub-criteria C4.1 and C4.2;
- Model 6: Calculation of the local values of the weight coefficients of the sub-criteria C5.1, C5.2 and C5.3.

**Step 1:** The ranking of the criteria and the sub-criteria was carried out according to the preferences of the experts (E1–E7). The defined ranks are shown in Table 7 using Expression (1).

**Step 2:** The comparative significance of the criteria and the sub-criteria was determined as shown in Table 7 using Expression (2) (as defined below). This significance was measured by applying triangular fuzzy numbers. At this stage, fuzzy comparisons by pairs of criteria were made according to the preferences of the experts via five linguistic terms, as shown in Table 8 [75].

**Table 7.** Linguistic evaluations of the criteria and the sub-criteria.

| | | C1–C5 | C1.1–C1.4 | C2.1–C2.2 | C3.1–C3.4 | C4.1–C4.2 | C5.1–C5.3 |
|---|---|---|---|---|---|---|---|
| E1 | R | C4 > C3 = C2 > C1 > C5 | C1.1 > C1.3 > C1.4 > C1.2 | C2.1 > C2.2 | C3.1 > C3.4 > C3.2 > C3.3 | C4.2 > C4.1 | C5.1 > C5.2 > C5.3 |
| | C | EI, AI, EI, VI, FI | EI, AI, FI, VI | EI, VI | EI, VI, EI, EI | EI, VI | EI, WI, AI |
| E2 | R | C1 > C5 > C3 > C2 > C4 | C1.4 > C1.1 > C1.2 > C1.3 | C2.1 > C2.2 | C3.4 > C3.1 > C3.3 > C3.2 | C4.2 > C4.1 | C5.2 > C5.1 > C5.3 |
| | C | EI, AI, FI, WI, WI | EI, FI, FI, WI | EI, WI | EI, AI, FI, WI | EI, WI | EI, VI, FI |
| E3 | R | C2 > C1 > C3 > C4 > C5 | C1.1 > C1.3 > C1.4 > C1.2 | C2.1 > C2.2 | C3.1 > C3.3 > C3.4 > C3.2 | C4.2 > C4.1 | C5.1 > C5.3 > C5.2 |
| | C | EI, EI, WI, FI, VI | EI, WI, EI, FI | EI, WI | EI, WI, WI, FI | EI, WI | EI, EI, WI |
| E4 | R | C1 > C5 > C4 > C3 > C2 | C1.1 > C1.4 > C1.2 > C1.3 | C2.2 > C2.1 | C3.1 > C3.3 > C3.2 > C3.4 | C4.2 > C4.1 | C5.1 > C5.2 = C5.3 |
| | C | EI, AI, VI, EI, VI | EI, AI, VI, WI | EI, EI | EI, AI, EI, WI | EI, AI | EI, AI, FI |
| E5 | R | C1 > C4 > C5 > C2 = C3 | C1.1 > C1.4 > C1.2 > C1.3 | C2.1 > C2.2 | C3.1 > C3.4 > C3.2 > C3.3 | C4.2 > C4.1 | C5.1 > C5.2 > C5.3 |
| | C | EI, VI, VI, WI, EI | EI, FI, VI, AI | EI, WI | EI, VI, WI, VI | EI, WI | EI, FI, AI |
| E6 | R | C4 > C1 > C3 > C2 > C5 | C1.1 > C1.4 > C1.3 > C1.2 | C2.1 > C2.2 | C3.1 > C3.4 > C3.2 > C3.3 | C4.1 > C4.2 | C5.1 > C5.3 > C5.2 |
| | C | EI, WI, FI, WI, FI | EI, WI, EI, WI | EI, FI | EI, WI, FI, EI | EI, WI | EI, FI, WI |
| E7 | R | C1 > C2 > C5 > C3 > C4 | C1.3 > C1.2 > C1.1 > C1.4 | C2.1 > C2.2 | C3.1 > C3.3 > C3.4 > C3.2 | C4.1 > C4.2 | C5.3 > C5.2 > C5.1 |
| | C | EI, VI, WI, WI, WI | EI, FI, FI, WI | EI, WI | EI, VI, WI, WI | EI, VI | EI, WI, EI |

R: Rank; C: Comparisons.

**Table 8.** Fuzzy language scale for criteria assessment [75].

| Linguistic Terms | Abbreviation | Fuzzy Number in a Triangular Style |
|---|---|---|
| Equally important | (EI) | (1, 1, 1) |
| Weakly important | (WI) | (2/3, 1, 3/2) |
| Fairly Important | (FI) | (3/2, 2, 5/2) |
| Very important | (VI) | (5/2, 3, 7/2) |
| Absolutely important | (AI) | (7/2, 4, 9/2) |

The vectors of the comparative significance were defined by calculating the comparative significance of the criteria and the sub-criteria (Appendix A). For example, the comparative significance of the criteria C1–C5, for expert E1, was obtained by applying Expression (2) as follows:

- $\varphi_{C4/C3} = \frac{\varpi_{C4}}{\varpi_{C3}} = \frac{AI}{EI} = (3.5; 4; 4.5)$;
- $\varphi_{C3/C2} = \frac{\varpi_{C3}}{\varpi_{C2}} = \frac{EI}{AI} = (0.22; 0.25; 0.29)$;
- $\varphi_{C2/C1} = \frac{\varpi_{C2}}{\varpi_{C1}} = \frac{VI}{EI} = (2.5; 3; 3.5)$;
- $\varphi_{C1/C5} = \frac{\varpi_{C1}}{\varpi_{C5}} = \frac{FI}{VI} = (0.43; 0.67; 1)$.

Thus, the vector of comparative significance was obtained by applying Expression (3):

$$\Phi^1 = ((3.5, 4, 4.5); (0.22, 0.25, 0.29); (2.5, 3, 3.5); (0.43, 0.67, 1)).$$

**Step 3:** In this step, the constraints of the fuzzy model were defined by applying Expression (4) and Expression (5).

The first group of the constraints of criteria C1–C5, for expert E1, was defined as follows: $\frac{\varpi_{C4}}{\varpi_{C3}} = (3.5; 4; 4.5)$, $\frac{\varpi_{C3}}{\varpi_{C2}} = (0.22; 0.25; 0.29)$, $\frac{\varpi_{C2}}{\varpi_{C1}} = (2.5; 3; 3.5)$ and $\frac{\varpi_{C1}}{\varpi_{C5}} = (0.43; 0.67; 1)$. The second group of the constraints resulting from the condition of transitivity of relations was defined as follows: $\frac{w_4}{w_2} = (3.5; 4.00; 4.5) * (0.1; 0.25; 0.28) =$

$(0.35; 1.00; 1.26)$; $\frac{w_3}{w_1} = (0.10; 0.25; 0.28) * (2.50; 3.00; 3.50) = (0.25; 0.75; 0.98)$ and $\frac{w_2}{w_5} = (2.50; 3.00; 3.50) * (0.43; 0.67; 1.00)$. The constraints of the other models were defined in the same way.

**Step 4:** On the basis of the constraints defined in the previous step, Model (6) was formed to determine the optimal fuzzy values of the weight coefficients of the criteria and the sub-criteria (Appendix B). It was used to specify the weight coefficients of the criteria and the sub-criteria, presented below.

Expert 1 $(C1 - C5)$ min $\chi$

s. t.

$$\left\{ \begin{array}{l} \left| \frac{w_4^l}{w_3^u} - 3.5 \right| \le \chi \; ; \left| \frac{w_4^m}{w_3^m} - 4 \right| \le \chi \; ; \left| \frac{w_4^u}{w_3^{ul}} - 4.5 \right| \le \chi; \\[2mm] \left| \frac{w_3^l}{w_2^u} - 0.22 \right| \le \chi \; ; \left| \frac{w_3^m}{w_2^m} - 0.25 \right| \le \chi \; ; \left| \frac{w_3^u}{w_2^l} - 0.29 \right| \le \chi; \\[2mm] \left| \frac{w_2^l}{w_1^u} - 2.5 \right| \le \chi \; ; \left| \frac{w_2^m}{w_1^m} - 3 \right| \le \chi \; ; \left| \frac{w_2^u}{w_1^l} - 3.5 \right| \le \chi; \\[2mm] \left| \frac{w_1^l}{w_5^u} - 0.43 \right| \le \chi \; ; \left| \frac{w_1^m}{w_3^m} - 0.67 \right| \le \chi \; ; \left| \frac{w_1^u}{w_3^l} - 1 \right| \le \chi; \\[2mm] \left| \frac{w_4^l}{w_2^u} - 0.78 \right| \le \chi \; ; \left| \frac{w_2^m}{w_4^m} - 1 \right| \le \chi \; ; \left| \frac{w_2^u}{w_4^l} - 1.29 \right| \le \chi; \quad \cdots \\[2mm] \left| \frac{w_3^l}{w_1^u} - 0.56 \right| \le \chi \; ; \left| \frac{w_3^m}{w_1^m} - 0.75 \right| \le \chi \; ; \left| \frac{w_3^u}{w_1^l} - 1.00 \right| \le \chi; \\[2mm] \left| \frac{w_2^l}{w_5^u} - 1.07 \right| \le \chi \; ; \left| \frac{w_2^m}{w_5^m} - 2.00 \right| \le \chi \; ; \left| \frac{w_2^u}{w_5^l} - 3.5 \right| \le \chi; \\[2mm] \sum_{j=1}^{5} (w_j^l + 4 \cdot w_j^m + w_j^u)/6 = 1 \; , \; \forall j = 1,2,3,4,5 \\[2mm] w_j^l \le w_j^m \le w_j^u, \; \forall j = 1,2,3,4,5 \\[1mm] w_j^l, w_j^m, w_j^u \ge 0 \; , \; \forall j = 1,2,3,4,5 \end{array} \right.$$

Expert 1 $(C5.1 - C5.3)$ min $\chi$

s. t.

$$\left\{ \begin{array}{l} \left| \frac{w_{5.1}^l}{w_{5.2}^u} - 0.67 \right| \le \chi \; ; \left| \frac{w_{5.1}^m}{w_{5.2}^m} - 1 \right| \le \chi \; ; \left| \frac{w_{5.1}^u}{w_{5.2}^l} - 1.5 \right| \le \chi; \\[2mm] \left| \frac{w_{5.2}^l}{w_{5.3}^u} - 2.33 \right| \le \chi \; ; \left| \frac{w_{5.2}^m}{w_{5.3}^m} - 4 \right| \le \chi \; ; \left| \frac{w_{5.2}^u}{w_{5.3}^l} - 6.7 \right| \le \chi; \\[2mm] \left| \frac{w_{5.1}^l}{w_{5.3}^u} - 1.56 \right| \le \chi \; ; \left| \frac{w_{5.1}^m}{w_{5.3}^m} - 4.00 \right| \le \chi \; ; \left| \frac{w_{5.1}^u}{w_{5.3}^l} - 10.07 \right| \le \chi; \\[4mm] \sum_{j=1}^{3} (w_j^l + 4 \cdot w_j^m + w_j^u)/6 = 1 \; , \; \forall j = 1,2,3, \\[4mm] w_j^l \le w_j^m \le w_j^u, \; \forall j = 1,2,3 \\[1mm] w_j^l, w_j^m, w_j^u \ge 0 \; , \; \forall j = 1,2,3 \end{array} \right.$$

**Step 5:** After solving the model, we obtained the optimal local values of the expert weight coefficients (Appendix C). The global sub-criteria values for each expert were obtained, as shown in Table 9, by multiplying the local sub-criteria values by the criteria weight coefficients. Lingo 17.0 software was utilized to solve the non-linear fuzzy models, which allowed for obtaining the mean value $\chi \approx 0.0$, showing the high consistency of the provided values of the criteria weights.

**Table 9.** Global values of the fuzzy weight coefficients of the sub-criteria.

| Criteria | Weight | Sub-Criteria | Local Weight | Global Weight |
|----------|--------|--------------|--------------|---------------|
| C1 | (0.2, 0.27, 0.27) | C1.1 | (0.2, 0.33, 0.35) | (0.04, 0.09, 0.1) |
|  |  | C1.2 | (0.1, 0.18, 0.19) | 0.02, 0.05, 0.05) |
|  |  | C1.3 | (0.15, 0.29, 0.34) | (0.03, 0.08, 0.09) |
|  |  | C1.4 | (0.17, 0.25, 0.27) | (0.03, 0.07, 0.07) |
| C2 | (0.16, 0.22, 0.23) | C2.1 | (0.051, 0.53, 0.73) | (0.08, 0.12, 0.17) |
|  |  | C2.2 | (0.41, 0.41, 0.58) | 0.07, 0.09, 0.13) |
| C3 | (0.13, 0.21, 0.23) | C3.1 | (0.18, 0.26, 0.37) | (0.02, 0.05, 0.08) |
|  |  | C3.2 | (0.17, 0.28, 0.33) | (0.02, 0.06, 0.08) |
|  |  | C3.3 | (0.16, 0.22, 0.23) | (0.02, 0.05, 0.05) |
|  |  | C3.4 | (0.17, 0.25, 0.28) | (0.02, 0.05, 0.06) |
| C4 | (0.11, 0.21, 0.26) | C4.1 | (0.43, 0.43, 0.59) | (0.05, 0.09, 0.15) |
|  |  | C4.2 | (0.52, 0.52, 0.67) | (0.06, 0.11, 0.18) |
| C5 | (0.08, 0.14, 0.17) | C5.1 | (0.35,0.4, 0.51) | (0.03, 0.05, 0.08) |
|  |  | C5.2 | (0.28, 0.31, 0.5) | (0.02, 0.04, 0.08) |
|  |  | C5.3 | (0.2, 0.24, 0.34) | (0.02, 0.03, 0.06) |

### 4.3. Ranking of Alternatives

After determining the weight of the criteria, compensatory (F-MAIRCA) and partially compensatory (F-PROMETHEE) approaches were used to select the best location of the logistics platform.

#### 4.3.1. The F-MAIRCA Results

Alternatives were ranked using Equations (8)–(16). The different steps of F-MAIRCA are described below.

Step 1: Individual evaluations of the alternatives were performed by experts (E1, E2, E3, E4, E5, E6 and E7) using the linguistic terms presented in Table 10 [76]. These evaluations are shown in Appendix D using Equation (8).

**Table 10.** Alternative language scale [76].

| Linguistic Term | Abbreviation | Fuzzy Number in a Triangular Style |
|---|---|---|
| Very Low | (VL) | (0, 1, 2) |
| Low | (L) | (1, 2, 3) |
| Medium | (M) | (2, 3, 4) |
| High | (H) | (3, 4, 5) |
| Very High | (VH) | (4, 5, 6) |

**Step 2:** the aggregated decision values of the triangular fuzzy numbers were calculated by Equation (9) to construct the fuzzy aggregated decision matrix, as shown in Table 11.

**Table 11.** Fuzzy aggregated decision matrix.

|  | A1 | A2 |  |  | A6 | A7 |
|---|---|---|---|---|---|---|
| C1.1 | (3.43, 4.43, 5.43) | (3.14, 4.14, 5.14) | ... | ... | (2.43, 3.43, 4.43) | (1.71, 2.71, 3.71) |
| C1.2 | (2.14, 3.00, 3.86) | (2.57, 3.57, 4.57) | ... | ... | (3.57, 4.57, 5.57) | (2.86, 3.86, 4.86) |
| ⋮ | ⋮ | ⋮ | ⋱ | ⋱ | ⋮ | ⋮ |
| C5.2 | (3.57, 4.57, 5.57) | (2.00, 3.00, 4.00) | ... | ... | (2.00, 3.00, 4.00) | (2.43, 3.43, 4.43) |
| C5.3 | (2.00, 3.00,4.00) | (2.14, 3.14, 4.14) | ... | ... | (1.29, 2.29, 3.29) | (2.00, 3.00, 4.00) |

**Step 3:** In this step, each alternative received equal preferences. In this study, we had seven alternatives. Thus, the preferences $PAi$ were calculated as follows: $PAi = 1/7 = 0.143$ by applying Equation (10).

**Step 4:** Using Equation (11), the fuzzy matrix of theoretical ponder $\widetilde{T}_{PA}$ was obtained as demonstrated in Table 12. Each row of this table indicates the theoretical evaluation of the alternatives for this particular criterion.

**Table 12.** Fuzzy matrix of theoretical ponder.

|  | A1 | A2 |  |  | A6 | A7 |
|---|---|---|---|---|---|---|
| C1.1 | (0.01, 0.01, 0.01) | (0.01, 0.01, 0.01) | ... | ... | (0.01, 0.01, 0.01) | (0.01, 0.01, 0.01) |
| C1.2 | (0.00, 0.01, 0.01) | (0.00, 0.01, 0.01) | ... | ... | (0.00, 0.01, 0.01) | (0.00, 0.01, 0.01) |
| ⋮ | ⋮ | ⋮ | ⋱ | ⋱ | ⋮ | ⋮ |
| C5.2 | (0.00 0.01, 0.01) | (0.00 0.01, 0.01) | ... | ... | (0.00 0.01, 0.01) | (0.00 0.01, 0.01) |
| C5.3 | (0.00, 0.00 0.01) | (0.00, 0.00 0.01) | ... | ... | (0.00, 0.00 0.01) | (0.00, 0.00 0.01) |

**Step 5:** To make the fuzzy aggregated decision matrix dimensionless, Equation (12) and Equation (13) were used. We then obtained the fuzzy normalized decision matrix, as shown in Table 13.

**Table 13.** Fuzzy Normalized Decision Matrix.

|  | **A1** | **A2** |  |  | **A6** | **A7** |
|---|---|---|---|---|---|---|
| C1.1 | (0.63, 0.82, 1.00) | (0.58, 0.76, 0.95) | ⋯ | ⋯ | (0.45, 0.63, 0.82) | (0.32, 0.50, 0.68) |
| C1.2 | (0.13, 0.38, 0.52) | (0.28, 0.48, 0.59) | ⋯ | ⋯ | (0.48, 0.59, 0.67) | (0.35, 0.52, 0.62) |
|  | ⋮ | ⋮ | ⋱ | ⋱ | ⋮ | ⋮ |
| C5.2 | (0.64, 0.82, 1.00) | (0.36, 0.54, 0.72) | ⋯ | ⋯ | (0.36, 0.54, 0.72) | (0.44, 0.62, 0.79) |
| C5.3 | (0.34, 0.59, 0.83) | (0.52, 0.76, 1.00) | ⋯ | ⋯ | (0.31, 0.55, 0.79) | (0.48, 0.72, 0.97) |

**Step 6:** Using Equation (14), the matrix of fuzzy actual ponder, also known as the real evaluation matrix, was provided, as shown in Table 14.

**Table 14.** Matrix of fuzzy actual ponder.

|  | **A1** | **A2** |  |  | **A6** | **A7** |
|---|---|---|---|---|---|---|
| C1.1 | (0.004, 0.010, 0.01) | (0.00, 0.010, 0.013) | ⋯ | ⋯ | (0.003, 0.008, 0.011) | (0.002, 0.006, 0.009) |
| C1.2 | (0.00, 0.004, 0.004) | (0.00, 0.003, 0.003) | ⋯ | ⋯ | (0.002, 0.003, 0.002) | (0.002, 0.003, 0.003) |
|  | ⋮ | ⋮ | ⋱ | ⋱ | ⋮ | ⋮ |
| C5.2 | (0.002, 0.005, 0.01) | (0.00, 0.003, 0.009) | ⋯ | ⋯ | (0.001, 0.003, 0.009) | (0.001, 0.004, 0.009) |
| C5.3 | (0.00, 0.003, 0.007) | (0.00, 0.004, 0.008) | ⋯ | ⋯ | (0.001, 0.003, 0.006) | (0.001, 0.003, 0.008) |

**Step 7:** In this step, the total gap matrix was calculated using Equation (15), as shown in Table 15.

**Table 15.** Total gap matrix.

|  | **A1** | **A2** | **A3** | **A4** | **A5** | **A6** | **A7** |
|---|---|---|---|---|---|---|---|
| C1.1 | 0.002 | 0.002 | 0.002 | 0.004 | 0.004 | 0.004 | 0.005 |
| C1.2 | 0.003 | 0.003 | 0.003 | 0.003 | 0.003 | 0.004 | 0.003 |
| C1.3 | 0.004 | 0.005 | 0.005 | 0.004 | 0.005 | 0.005 | 0.005 |
| C1.4 | 0.006 | 0.008 | 0.008 | 0.007 | 0.007 | 0.007 | 0.008 |
| C2.1 | 0.003 | 0.004 | 0.004 | 0.004 | 0.004 | 0.005 | 0.003 |
| C2.2 | 0.006 | 0.006 | 0.006 | 0.007 | 0.007 | 0.007 | 0.008 |
| C3.1 | 0.001 | 0.001 | 0.001 | 0.002 | 0.002 | 0.002 | 0.001 |
| C3.2 | 0.004 | 0.005 | 0.005 | 0.005 | 0.005 | 0.006 | 0.005 |
| C3.3 | 0.004 | 0.004 | 0.004 | 0.004 | 0.004 | 0.004 | 0.004 |
| C3.4 | 0.006 | 0.006 | 0.006 | 0.007 | 0.007 | 0.007 | 0.007 |
| C4.1 | 0.003 | 0.005 | 0.005 | 0.005 | 0.006 | 0.006 | 0.005 |
| C4.2 | 0.003 | 0.007 | 0.007 | 0.007 | 0.007 | 0.007 | 0.005 |
| C5.1 | 0.001 | 0.003 | 0.003 | 0.003 | 0.003 | 0.003 | 0.006 |
| C5.2 | 0.001 | 0.003 | 0.003 | 0.003 | 0.003 | 0.003 | 0.002 |
| C5.3 | 0.002 | 0.001 | 0.001 | 0.002 | 0.002 | 0.002 | 0.001 |

**Step 8:** Table 16 presents the gap values as given in Equation (16). The aim of the decision-makers is to maintain the smallest possible value between the theoretical and actual evaluation for the best alternative, according to the FMAIRCA method. The final ranking of the gap values in ascending order is as follows: A1 > A3 > A2 > A6 > A4 > A7 > A5.

**Table 16.** Ranking of alternatives.

| | Alternatives | | | | | | |
|---|---|---|---|---|---|---|---|
| | A1 | A2 | A3 | A4 | A5 | A6 | A7 |
| Gap values | 0.048 | 0.064 | 0.063 | 0.068 | 0.067 | 0.070 | 0.068 |
| Rank | 1 | 3 | 2 | 6 | 4 | 7 | 5 |

### 4.3.2. The F-PROMETHEE Results

**Steps 1 and 2:** The individual evaluations of the alternatives according to different criteria and the aggregated decision matrix are shown in Tables 11 and 12.

**Steps 3, 4 and 5:** The distances between the two alternatives with respect to each criterion were calculated by applying Equation (17). After that, the distances were expressed as a preference function for every pair of alternatives. In this step, the usual criterion preference function was used, and the results were summarized, as shown in Appendix E. The preference index for each pair of alternatives was determined using Equation (18). For example, the preference of alternative A1 over A2 was represented by the index PC1.1 (A1, A2) = (0.2, 0.33, 0.35). The fuzzy preferences index was obtained, as shown in Appendix E, using the value preference index for each criterion and the weights of the criteria. For simplicity, Appendix E presents only the results corresponding to the values of the usual criterion preference function and the overall preference index.

**Steps 6, 7 and 8:** The defuzzified values of leaving flows, entering flows and net flows were respectively calculated, as shown in Table 17. The ranking results indicated that alternative A1 was the most durable location (with scores of 0.981, 1.30 and 0.317 for Ø net, Ø + and Ø, respectively), while alternative A3 was the second-best sustainable location and alternative A6 was the least sustainable location.

**Table 17.** Defuzzified upgrading flows.

| | A1 | A2 | A3 | A4 | A5 | A6 | A7 |
|---|---|---|---|---|---|---|---|
| $\Phi^+$ | 1.30 | 1.30 | 1.75 | 0.91 | 1.37 | 0.60 | 1.22 |
| $\Phi-$ | 0.317 | 0.792 | 0.899 | 1.55 | 1.385 | 2.074 | 1.43 |
| $\Phi$net | 0.981 | 0.506 | 0.851 | −0.64 | −0.01 | −1.47 | −0.21 |
| Rank | 1 | 3 | 2 | 6 | 4 | 7 | 5 |

### *4.4. Stability of the Obtained Results*

4.4.1. Assessment of the Independence of the Aggregation Technique

In Figure 3, the ranking of alternatives was sorted by the F-PROMETHEE, F-MAIRCA and F-TOPSIS methods with the partial compensation technique, compensation technique and non-compensation technique, respectively. The obtained results prove that the use of the non-compensation technique produced a different ranking. As proposed in this paper, the compensatory and partially compensatory composite indicators can be used by decision-makers to locate the logistics platforms.

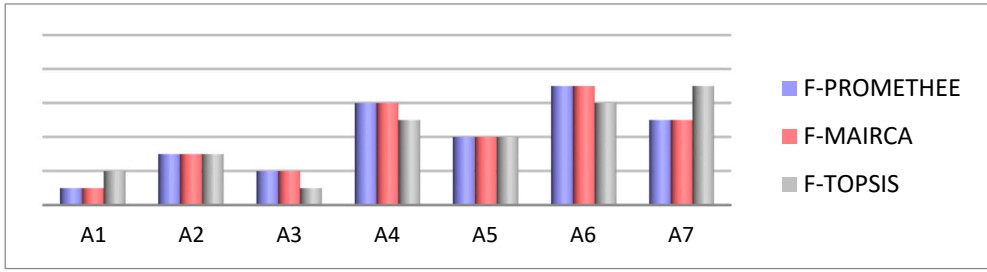

**Figure 3.** Ranking of alternatives for the three aggregation technique.

### 4.4.2. Variation of Criteria Weights

The illustrative ranking of F-MAIRCA and F-PROMETHEE changed by varying the applied scenarios of weight sensitivity analysis, as shown in Figure 4. Obviously, the variation of the weights of the criteria through the scenarios changed the ranking of the alternatives in both methods. Despite these changes, a logical ranking was obtained. The best and worse alternative's positions were almost identical to the initial ranking. The obtained results prove that the proposed approach is slightly sensitive to the variations in the criteria weights.

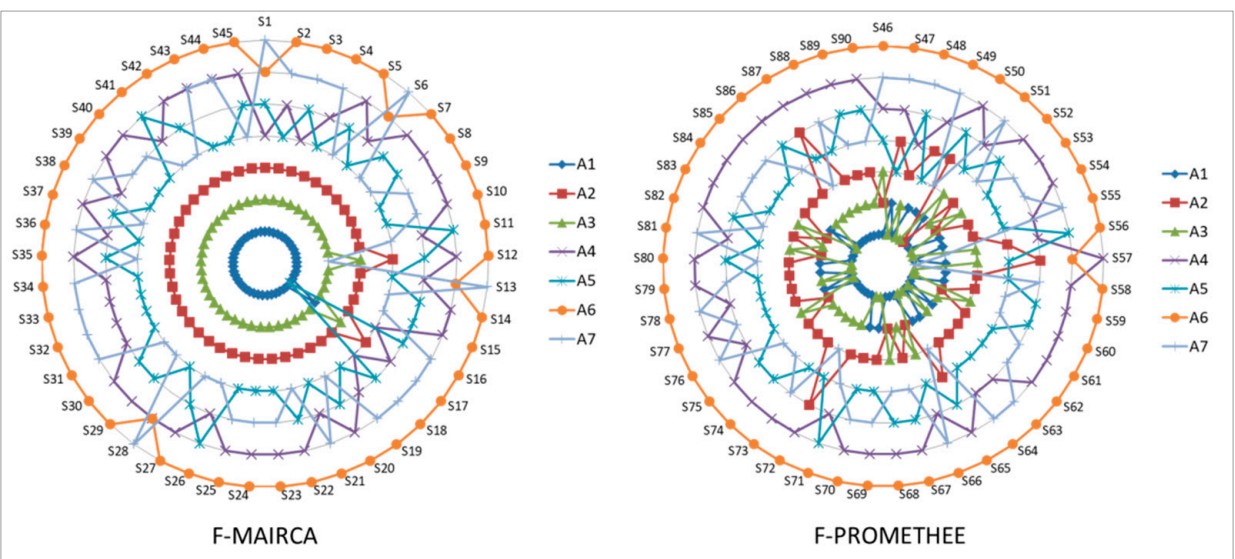

**Figure 4.** The illustrative ranking of the fuzzy multi-attribute ideal-real comparative analysis (F-MAIRCA) and the fuzzy preference ranking organization method for enrichment evaluation (F-PROMETHEE) obtained by sensitivity analysis.

### 4.5. Results and Decision-Making Process

The F-FUCOM results revealed the following order: economic criteria > environmental criteria > political criteria > social criteria > territorial criteria. From this order, we noticed that economic sustainability was the most important aspect to be taken into account to evaluate the logistics platform. The next two most important aspects were environmental sustainability and political sustainability.

Figure 5 represents the ranking of the sustainability sub-criteria (indicator) of each criterion (dimension). This figure shows that the conformity with the environmental emissions regulations is ranked one, and were respectively followed by (C4.2) the role of support to the industry and (4.1) the current policy. The effect on the natural landscape, connectivity to multimodal transport and fiscal policies were the most important sub-criteria to locate the logistics platform in the fourth, fifth and sixth place, respectively.

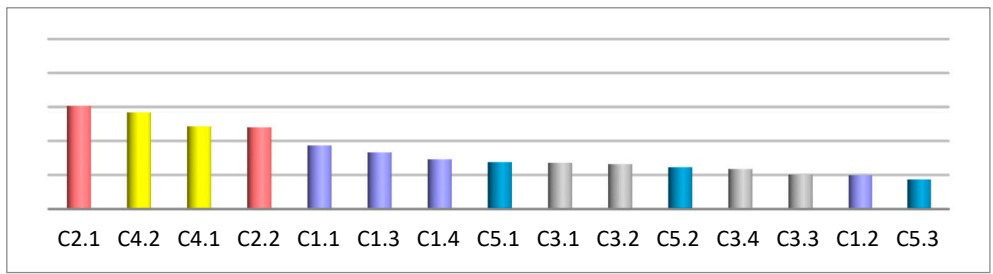

**Figure 5.** Results obtained by applying the full consistency method (FUCOM)—weighting of the criteria.

- The ranking of the economic criteria was as follows: C1.1 > C1.3 > C1.4 > C1.2. The above results show that a location should ensure connectivity to multimodal transport and offer fiscal policies to attract investors and promote the development of multimodal transport.
- The ranking of the environmental criteria was in the following order: C2.1> C2.2. The conformity with environmental emissions regulations was at the top of the list, which was expected because the improvement of environmental criteria is important in the process of the logistics platform localization.
- The ranking of the social criteria was as follows: C3.1 > C3.2 > C3.4 > C3.3. The results presented above reveal that the logistics platform should ensure the safety and security of the site and the workers, while minimizing the generated noise.
- The results of ranking the political criteria showed the following order: C4.2 > C4.1, which proves the vital role of support and cooperation between both government and industry in choosing the platform location, as locations are often not finalized due to government instability.
- The ranking of the territorial criteria was as follows: C5.1 > C5.2 > C5.3. The above ranking order demonstrates the importance of a location being connected to and accessible by all transport modes. Second, a logistics platform should be close to all industrial areas.

The objective of this study was to develop a multi-criteria approach based on compensation phenomenon to locate the logistics platform by taking sustainability into account. The developed approach was applied to evaluate the influence of both aggregation techniques (compensatory and partially compensatory) on the final results. The findings of this case study showed that the two fuzzy MCDM (F-PROMETHEE and F-MAIRCA) methods provided similar location ranking results (A1 > A3 > A2 > A6 > A4 > A7 > A5) (Figure 6).

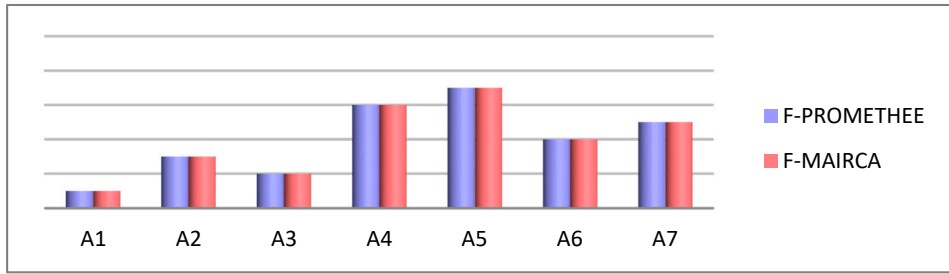

**Figure 6.** Final ranking of F-PROMETHEE and F-MAIRCA.

The authors can thus conclude that the choice of one of these methods seems appropriate for the selection of the most sustainable location. The purpose of this case study was not to compare the F-PROMETHEE and F-MAIRCA methods, but to highlight the capacity of the proposed approach to allow for the use of both aggregation techniques.

## 5. Implications

This study provides fuzzy compensatory and partially compensatory composite indicators to locate a logistics platform under sustainability perspectives. In this context, the main contributions of this study can be summed up in four points:

- First, in contrast to the existing localization approaches, in addition to the classic dimensions of sustainability (economic, environmental and social), this study included two other dimensions (political and territorial) identified in the literature as being relevant to urban logistics;
- Second, the proposed approach is characterized by the choice of methods that are most suitable to our study context. Although several MCDM methods were proposed, the decision-maker faces many the challenges when selecting the appropriate method to use to avoid the subjective choice. Thus, this study was carried out to manage the complexity of the decision-making process in situations of uncertainty. From a

methodological point of view, the present work integrates the set of fuzzy numbers with FUCOM, MAIRCA and PROMETHEE to locate the logistics platform. These methods were chosen because of their popularity and stability;

- Third, in the literature, there is no general or systematic localization method specifically related to the sustainability perspectives. This study proposes an innovative and interesting approach as a support tool for decision-makers with sustainability perspectives. The novelty of this approach lays in developing fuzzy compensatory and partially compensatory composite indicators by considering weak sustainability and limited sustainability;
- Fourth, to validate the robustness of the proposed approach, a sensitivity analysis was performed. In the first phase, the independence of the aggregation technique was assessed. However, in the second phase, the effect of sensitivity on the variation of the criteria weight was evaluated.

From a practical perspective, this research provides several results for sustainable facility location problems. The obtained findings provide valuable insights for decision-makers to select a logistic platform from the calculated composite indicators.

## 6. Conclusions

An approach for locating the logistics platforms with sustainability perspectives was introduced in this paper. It uses an integrated MCDM method with the set of fuzzy numbers. A composite indicator based on compensatory and partially compensatory multi-criteria decision-making methods was also proposed. The composite indicator computation model involves two stages. Firstly, important weights of sustainability criteria were computed using the F-FUCOM method, relying on the experts' linguistic responses. Secondly, alternatives were classified by two aggregation methods: F-MAIRCA and F-PROMETHEE. The suggested approach was applied in the city of Sfax, where it is necessary to construct a logistics platform to reduce the impact of freight transport in the city.

The experimental results reveal that the economic and environmental criteria considerably affect the selection of the logistics platform location. A good connection and accessibility to multimodal transport is essential for a platform's success. The chosen location must ensure conformity with the environmental emissions regulations. Furthermore, the logistics platform should ensure the safety and security of the site and the workers, while minimizing the generated noise. A vital role of support and cooperation between both the government and industry is needed to choose the adequate platform location.

From a practical perspective, this research proposed a solution to sustainable facility location problems. In fact, one of our main contributions is to present a comprehensive list of sustainable evaluation sub-criteria, involving economic, environmental, social, political and territorial criterion, to assess the sustainability of the logistics platform location. The novelty of this approach lays in developing fuzzy compensatory and partially compensatory composite indicators to solve the facility location problem. Then, the stability and the robustness of the proposed approach were demonstrated.

However, this approach has some limitations. Firstly, it used only a limited number of experts. Secondly, the MCDM methods applied in this research work rely on the experts' opinions, which can deteriorate their performances. Finally, this study only considers uncertainty with the fuzzy set theory. Limited efforts were made to simultaneously relate two types of uncertainty with MCDMs to solve decision-making problems [77].

The authors suggest several future research directions. In further research, it is recommended to integrate fuzzy theory with other types of uncertainty (such as stochastic). We expect to build ontologies from the sustainability criteria and the stakeholders of the logistics platform to support the proposed approach. Furthermore, we will try to improve our sensitivity analysis by using other models, methods and tools.

**Author Contributions:** Conceptualization, H.A. and N.H.; methodology, H.A., N.H., M.B. and L.K.; software, H.A.; validation, H.A., N.H., M.B. and L.K.; formal analysis, H.A.; investigation, H.A.; data curation, H.A.; writing—original draft preparation, H.A.; writing—review and editing, H.A., N.H., M.B. and L.K. All authors have read and agreed to the published version of the manuscript.

**Funding:** This research received no external funding.

**Institutional Review Board Statement:** Not applicable.

**Informed Consent Statement:** Not applicable.

**Data Availability Statement:** Not applicable.

**Acknowledgments:** This work could have not been carried out without the generous collaboration of anonymous experts from different fields and disciplines.

**Conflicts of Interest:** The authors declare no conflict of interest.

## Appendix A

**Table A1.** Vectors of comparative meanings of criteria and sub-criteria.

| | Expert 1 (E1) | Expert 2 (E2) |
|---|---|---|
| C1–C5 | (3.5, 4, 4.5) (0.22, 0.25, 0.29) (2.5, 3, 3.5) (0.43, 0.67, 1) | (3.5, 4, 4.5) (0.33, 0.5, 0.71) (0.27, 0.5, 1) (0.45, 1, 2.24) |
| C1.1–C1.4 | (3.5, 4, 4.5) (0.33, 0.5, 0.71) (1, 1.5, 2.33) | (1.5, 2, 2.5) (0.6, 1, 1.67) (0.27, 0.5, 1) |
| C2.1–C2.2 | (0.67, 1, 1.5) | (0.67, 1, 1.5) |
| C3.1–C3.4 | (2.5, 3, 3.5) (0.29, 0.33, 0.4) (1, 1, 1) | (3.5, 4, 4.5) (0.33, 0.5, 0.71) (0.27, 0.5, 1) |
| C4.1–C4.2 | (2.5, 3, 3.5) | (0.67, 1, 1.5) |
| C5.1–C5.3 | (0.67, 1, 1.5) (2.33, 4, 6.72) | (2.5, 3, 3.5) (0.43, 0.67, 1) |
| | **Expert 5 (E5)** | **Expert 6 (E6)** |
| C1–C5 | (1, 1, 1) (0.67, 1, 1.5) (1, 2, 3.73) (1, 1.5, 2.33) | (3.5, 4, 4.5) (0.56, 0.75, 1) (0.29, 0.33, 0.4) (2.5, 3, 3.5) |
| C1.1–C1.4 | (0.67, 1, 1.5) (0.67, 1, 1.49) (1.5, 2, 2.5) | (3.5, 4, 4.5) (0.56, 0.75, 1) (0.19, 0.33, 0.6) |
| C2.1–C2.2 | (0.67, 1, 1.5) | (1, 1, 1) |
| C3.1–C3.4 | (0.67, 1, 1.5) (0.45, 1, 2.24) (1, 2, 3.73) | (3.5, 4, 4.5) (0.22, 0.25, 0.29) (0.67, 1, 1.5) |
| C4.1–C4.2 | (0.67, 1, 1.5) | (3.5, 4, 4.5) |
| C5.1–C5.3 | (1, 1, 1) (0.67, 1, 1.5) | (3.5, 4, 4.5) (0.33, 0.5, 0.71) |
| | **Expert 5 (E5)** | **Expert 6 (E6)** |
| C1–C5 | (2.5, 3, 3.5) (0.71, 1, 1.4) (0.19, 0.33, 0.6) (0.67, 1, 1.49) | (0.67, 1, 1.5) (1, 2, 3.73) (0.27, 0.5, 1) (1, 2, 3.73) |
| C1.1–C1.4 | (1.5, 2, 2.5) (1, 1.5, 2.33) (1, 1.33, 1.8) | (0.67, 1, 1.5) (0.67, 1, 1.49) (0.67, 1, 1.5) |
| C2.1–C2.2 | (0.67, 1, 1.5) | (1.5, 2, 2.5) |
| C3.1–C3.4 | (2.5, 3, 3.5) (0.19, 0.33, 0.6) (1.67, 3, 5.22) | (0.67, 1, 1.5) (1, 2, 3.73) (0.4, 0.5 0.67) |
| C4.1–C4.2 | (0.67, 1, 1.5) | (0.67, 1, 1.5) |
| C5.1–C5.3 | (1.5, 2, 2.5) (1.4, 2, 3) | (1.5, 2, 2.5) (0.27, 0.5, 1) |
| | **Expert 7 (E7)** | |
| C1–C5 | (2.5, 3, 3.5) (0.19, 0.33, 0.6) (0.45, 1, 2.24) (0.45, 1, 2.24) | |
| C1.1–C1.4 | (1.5, 2, 2.5) (0.6, 1, 1.67) (0.27, 0.5, 1) | |
| C2.1–C2.2 | (0.67, 1, 1.5) | |
| C3.1–C3.4 | (2.5, 3, 3.5) (0.19, 0.33, 0.6) (0.45, 1, 2.24) | |
| C4.1–C4.2 | (2.5, 3, 3.5) | |
| C5.1–C5.3 | (0.67, 1, 1.5) (0.67, 1, 1.49) | |

# Appendix B

$$\text{Expert } \mathbf{1} \ (C1 - C5) \ \min \chi$$
$$\text{s.t.}$$

$$\left\{
\begin{array}{l}
\left|\dfrac{w_4^l}{w_3^u} - 3.5\right| \le \chi \ ; \ \left|\dfrac{w_4^m}{w_3^m} - 4\right| \le \chi \ ; \ \left|\dfrac{w_4^u}{w_3^{ul}} - 4.5\right| \le \chi; \\[10pt]
\left|\dfrac{w_3^l}{w_2^u} - 0.22\right| \le \chi \ ; \ \left|\dfrac{w_3^m}{w_2^m} - 0.25\right| \le \chi \ ; \ \left|\dfrac{w_3^u}{w_2^l} - 0.29\right| \le \chi; \\[10pt]
\left|\dfrac{w_2^l}{w_1^u} - 2.5\right| \le \chi \ ; \ \left|\dfrac{w_2^m}{w_1^m} - 3\right| \le \chi \ ; \ \left|\dfrac{w_2^u}{w_1^l} - 3.5\right| \le \chi; \\[10pt]
\left|\dfrac{w_1^l}{w_5^u} - 0.43\right| \le \chi \ ; \ \left|\dfrac{w_1^m}{w_3^m} - 0.67\right| \le \chi \ ; \ \left|\dfrac{w_1^u}{w_3^l} - 1\right| \le \chi; \\[10pt]
\left|\dfrac{w_4^l}{w_2^u} - 0.78\right| \le \chi \ ; \ \left|\dfrac{w_2^m}{w_4^m} - 1\right| \le \chi \ ; \ \left|\dfrac{w_2^u}{w_4^l} - 1.29\right| \le \chi; \quad \cdots\cdots \\[10pt]
\left|\dfrac{w_3^l}{w_1^u} - 0.56\right| \le \chi \ ; \ \left|\dfrac{w_3^m}{w_1^m} - 0.75\right| \le \chi \ ; \ \left|\dfrac{w_3^u}{w_1^l} - 1.00\right| \le \chi; \\[10pt]
\left|\dfrac{w_2^l}{w_5^u} - 1.07\right| \le \chi \ ; \ \left|\dfrac{w_2^m}{w_5^m} - 2.00\right| \le \chi \ ; \ \left|\dfrac{w_2^u}{w_5^l} - 3.5\right| \le \chi; \\[10pt]
\displaystyle\sum_{j=1}^{5}(w_j^l + 4 \cdot w_j^m + w_j^u)/6 = 1 \ , \ \forall j = 1,2,3,4,5 \\[10pt]
\qquad w_j^l \le w_j^m \le w_j^u, \ \forall j = 1,2,3,4,5 \\[6pt]
\qquad w_j^l, w_j^m, w_j^u \ge 0 \ , \ \forall j = 1,2,3,4,5
\end{array}
\right.$$

$$\text{Expert } \mathbf{7} \ (C1 - C5) \ \min \chi$$
$$\text{s.t.}$$

$$\left\{
\begin{array}{l}
\left|\dfrac{w_1^l}{w_2^u} - 2.5\right| \le \chi \ ; \ \left|\dfrac{w_1^m}{w_2^m} - 3\right| \le \chi \ ; \ \left|\dfrac{w_1^u}{w_2^{ul}} - 3.5\right| \le \chi; \\[10pt]
\left|\dfrac{w_2^l}{w_5^u} - 0.19\right| \le \chi \ ; \ \left|\dfrac{w_2^m}{w_5^m} - 0.33\right| \le \chi \ ; \ \left|\dfrac{w_2^u}{w_5^l} - 0.6\right| \le \chi; \\[10pt]
\left|\dfrac{w_5^l}{w_3^u} - 0.45\right| \le \chi \ ; \ \left|\dfrac{w_5^m}{w_3^m} - 1\right| \le \chi \ ; \ \left|\dfrac{w_5^u}{w_3^l} - 2.24\right| \le \chi; \\[10pt]
\left|\dfrac{w_3^l}{w_4^u} - 0.45\right| \le \chi \ ; \ \left|\dfrac{w_3^m}{w_4^m} - 1\right| \le \chi \ ; \ \left|\dfrac{w_3^u}{w_4^l} - 2.24\right| \le \chi; \\[10pt]
\left|\dfrac{w_1^l}{w_5^u} - 0.48\right| \le \chi \ ; \ \left|\dfrac{w_1^m}{w_5^m} - 1\right| \le \chi \ ; \ \left|\dfrac{w_1^u}{w_5^l} - 2.1\right| \le \chi; \\[10pt]
\left|\dfrac{w_2^l}{w_3^u} - 0.09\right| \le \chi \ ; \ \left|\dfrac{w_2^m}{w_3^m} - 0.33\right| \le \chi \ ; \ \left|\dfrac{w_2^u}{w_3^l} - 1.34\right| \le \chi; \\[10pt]
\left|\dfrac{w_5^l}{w_4^u} - 0.2\right| \le \chi \ ; \ \left|\dfrac{w_5^m}{w_4^m} - 1\right| \le \chi \ ; \ \left|\dfrac{w_5^u}{w_4^l} - 5.01\right| \le \chi; \\[10pt]
\displaystyle\sum_{j=1}^{5}(w_j^l + 4 \cdot w_j^m + w_j^u)/6 = 1 \ , \ \forall j = 1,2,3,4,5 \\[10pt]
\qquad w_j^l \le w_j^m \le w_j^u, \ \forall j = 1,2,3,4,5 \\[6pt]
\qquad w_j^l, w_j^m, w_j^u \ge 0 \ , \ \forall j = 1,2,3,4,5
\end{array}
\right.$$

$$\text{Expert } 1 \ (C1.1 - C1.4) \ \min \chi$$
$$\text{s.t.}$$

$$\left\{
\begin{array}{l}
\left|\dfrac{w_{1.1}^l}{w_{1.3}^u} - 3.5\right| \le \chi \ ; \ \left|\dfrac{w_{1.1}^m}{w_{1.3}^m} - 4\right| \le \chi \ ; \ \left|\dfrac{w_{1.1}^u}{w_{1.3}^l} - 4.5\right| \le \chi; \\[10pt]
\left|\dfrac{w_{1.3}^l}{w_{1.4}^u} - 0.33\right| \le \chi \ ; \ \left|\dfrac{w_{1.3}^m}{w_{1.4}^m} - 0.5\right| \le \chi \ ; \ \left|\dfrac{w_{1.3}^u}{w_{1.4}^l} - 0.71\right| \le \chi; \\[10pt]
\left|\dfrac{w_{1.4}^l}{w_{1.2}^u} - 1.00\right| \le \chi \ ; \ \left|\dfrac{w_{1.4}^m}{w_{1.2}^m} - 1.5\right| \le \chi \ ; \ \left|\dfrac{w_{1.4}^u}{w_{1.2}^l} - 2.33\right| \le \chi; \\[10pt]
\left|\dfrac{w_{1.1}^l}{w_{1.4}^u} - 1.17\right| \le \chi \ ; \ \left|\dfrac{w_{1.1}^m}{w_{1.4}^m} - 2.00\right| \le \chi \ ; \ \left|\dfrac{w_{1.1}^u}{w_{1.4}^l} - 3.21\right| \le \chi; \quad \cdots\cdots \\[10pt]
\left|\dfrac{w_{1.3}^l}{w_{1.2}^u} - 0.33\right| \le \chi \ ; \ \left|\dfrac{w_{1.3}^m}{w_{1.2}^m} - 0.75\right| \le \chi \ ; \ \left|\dfrac{w_{1.3}^u}{w_{1.2}^l} - 1.67\right| \le \chi; \\[10pt]
\displaystyle\sum_{j=1}^{4}(w_j^l + 4 \cdot w_j^m + w_j^u)/6 = 1 \ , \ \forall j = 1,2,3,4 \\[10pt]
\qquad w_j^l \le w_j^m \le w_j^u, \ \forall j = 1,2,3,4 \\[6pt]
\qquad w_j^l, w_j^m, w_j^u \ge 0 \ , \ \forall j = 1,2,3,4
\end{array}
\right.$$

$$\text{Expert } 7 \ (C1.1 - C1.4) \ \min \chi$$
$$\text{s.t.}$$

$$\left\{
\begin{array}{l}
\left|\dfrac{w_{1.3}^l}{w_{1.2}^u} - 1.5\right| \le \chi \ ; \ \left|\dfrac{w_{1.3}^m}{w_{1.2}^m} - 2\right| \le \chi \ ; \ \left|\dfrac{w_{1.3}^u}{w_{1.2}^l} - 2.5\right| \le \chi; \\[10pt]
\left|\dfrac{w_{1.2}^l}{w_{1.1}^u} - 0.6\right| \le \chi \ ; \ \left|\dfrac{w_{1.2}^m}{w_{1.1}^m} - 1\right| \le \chi \ ; \ \left|\dfrac{w_{1.2}^u}{w_{1.1}^l} - 1.67\right| \le \chi; \\[10pt]
\left|\dfrac{w_{1.1}^l}{w_{1.4}^u} - 0.27\right| \le \chi \ ; \ \left|\dfrac{w_{1.1}^m}{w_{1.4}^m} - 0.5\right| \le \chi \ ; \ \left|\dfrac{w_{1.1}^u}{w_{1.4}^l} - 1\right| \le \chi; \\[10pt]
\left|\dfrac{w_{1.3}^l}{w_{1.1}^u} - 0.9\right| \le \chi \ ; \ \left|\dfrac{w_{1.3}^m}{w_{1.1}^m} - 2\right| \le \chi \ ; \ \left|\dfrac{w_{1.3}^u}{w_{1.1}^l} - 4.17\right| \le \chi; \\[10pt]
\left|\dfrac{w_{1.2}^l}{w_{1.4}^u} - 0.16\right| \le \chi \ ; \ \left|\dfrac{w_{1.2}^m}{w_{1.4}^m} - 0.5\right| \le \chi \ ; \ \left|\dfrac{w_{1.2}^u}{w_{1.4}^l} - 1.67\right| \le \chi; \\[10pt]
\displaystyle\sum_{j=1}^{4}(w_j^l + 4 \cdot w_j^m + w_j^u)/6 = 1 \ , \ \forall j = 1,2,3,4 \\[10pt]
\qquad w_j^l \le w_j^m \le w_j^u, \ \forall j = 1,2,3,4 \\[6pt]
\qquad w_j^l, w_j^m, w_j^u \ge 0 \ , \ \forall j = 1,2,3,4
\end{array}
\right.$$

$$\text{Expert } 1 \ (C2.1 - C2.2) \ \min \chi$$
$$\text{s.t.}$$

$$\left\{
\begin{array}{l}
\left|\dfrac{w_{2.1}^l}{w_{2.2}^u} - 2.5\right| \le \chi \ ; \ \left|\dfrac{w_{2.1}^m}{w_{2.2}^m} - 3\right| \le \chi \ ; \ \left|\dfrac{w_{2.1}^u}{w_{2.2}^l} - 3.5\right| \le \chi; \\[10pt]
(w_{2.1}^l + 4 \cdot w_{2.1}^m + w_{2.1}^u)/6 + (w_{2.2}^l + 4 \cdot w_{2.2}^m + w_{2.2}^u)/6 = 1 \\[6pt]
\qquad w_j^l \le w_j^m \le w_j^u, \ \forall j = 1,2 \\[6pt]
\qquad w_j^l, w_j^m, w_j^u \ge 0 \ , \ \forall j = 1,2
\end{array}
\right. \quad \cdots\cdots$$

$$\text{Expert } 7 \ (C2.1 - C2.2) \ \min \chi$$
$$\text{s.t.}$$

$$\left\{
\begin{array}{l}
\left|\dfrac{w_{2.1}^l}{w_{2.2}^u} - 0.67\right| \le \chi \ ; \ \left|\dfrac{w_{2.1}^m}{w_{2.2}^m} - 1\right| \le \chi \ ; \ \left|\dfrac{w_{2.1}^u}{w_{2.2}^l} - 1.5\right| \le \chi; \\[10pt]
(w_{2.1}^l + 4 \cdot w_{2.1}^m + w_{2.1}^u)/6 + (w_{2.2}^l + 4 \cdot w_{2.2}^m + w_{2.2}^u)/6 = \\[6pt]
\qquad w_j^l \le w_j^m \le w_j^u, \ \forall j = 1,2 \\[6pt]
\qquad w_j^l, w_j^m, w_j^u \ge 0 \ , \ \forall j = 1,2
\end{array}
\right.$$

**Figure A1.** *Cont.*

Expert 1　$(C3.1 - C3.4)$ min $\chi$
s.t.

$$
\begin{cases}
\left|\frac{w_{3.1}^l}{w_{3.4}^u} - 2.5\right| \le \chi \; ; \left|\frac{w_{3.1}^m}{w_{3.4}^m} - 3\right| \le \chi \; ; \left|\frac{w_{3.1}^u}{w_{3.4}^l} - 3.5\right| \le \chi; \\[2mm]
\left|\frac{w_{3.4}^l}{w_{3.2}^u} - 0.29\right| \le \chi \; ; \left|\frac{w_{3.4}^m}{w_{3.2}^m} - 0.33\right| \le \chi \; ; \left|\frac{w_{3.4}^u}{w_{3.2}^l} - 0.4\right| \le \chi; \\[2mm]
\left|\frac{w_{3.2}^l}{w_{3.3}^u} - 1\right| \le \chi \; ; \left|\frac{w_{3.2}^m}{w_{3.3}^m} - 1\right| \le \chi \; ; \left|\frac{w_{3.2}^u}{w_{3.3}^l} - 1\right| \le \chi; \\[2mm]
\left|\frac{w_{3.1}^l}{w_{3.2}^u} - 0.7\right| \le \chi \; ; \left|\frac{w_{3.1}^m}{w_{3.2}^m} - 0.1\right| \le \chi \; ; \left|\frac{w_{3.1}^u}{w_{3.2}^l} - 1.4\right| \le \chi; \quad \cdots\cdots \\[2mm]
\left|\frac{w_{3.4}^l}{w_{3.3}^u} - 0.29\right| \le \chi \; ; \left|\frac{w_{3.4}^m}{w_{3.3}^m} - 0.33\right| \le \chi \; ; \left|\frac{w_{3.4}^u}{w_{3.3}^l} - 0.4\right| \le \chi; \\[2mm]
\sum_{j=1}^{4}(w_j^l + 4 \cdot w_j^m + w_j^u)/6 = 1 \; , \forall j = 1,2,3,4 \\[2mm]
w_j^l \le w_j^m \le w_j^u, \forall j = 1,2,3,4 \\[1mm]
w_j^l, w_j^m, w_j^u \ge 0 \; , \forall j = 1,2,3,4
\end{cases}
$$

Expert 7　$(C3.1 - C3.4)$ min $\chi$
s.t.

$$
\begin{cases}
\left|\frac{w_{3.1}^l}{w_{3.3}^u} - 2.5\right| \le \chi \; ; \left|\frac{w_{3.1}^m}{w_{3.3}^m} - 3\right| \le \chi \; ; \left|\frac{w_{3.1}^u}{w_{3.3}^l} - 3.5\right| \le \chi; \\[2mm]
\left|\frac{w_{3.3}^l}{w_{3.4}^u} - 0.19\right| \le \chi \; ; \left|\frac{w_{3.3}^m}{w_{3.4}^m} - 0.33\right| \le \chi \; ; \left|\frac{w_{3.3}^u}{w_{3.4}^l} - 0.6\right| \le \chi; \\[2mm]
\left|\frac{w_{3.4}^l}{w_{3.2}^u} - 0.45\right| \le \chi \; ; \left|\frac{w_{3.4}^m}{w_{3.2}^m} - 1\right| \le \chi \; ; \left|\frac{w_{3.4}^u}{w_{3.2}^l} - 2.24\right| \le \chi; \\[2mm]
\left|\frac{w_{3.1}^l}{w_{3.4}^u} - 0.48\right| \le \chi \; ; \left|\frac{w_{3.1}^m}{w_{3.4}^m} - 1\right| \le \chi \; ; \left|\frac{w_{3.1}^u}{w_{3.4}^l} - 2.1\right| \le \chi; \\[2mm]
\left|\frac{w_{3.3}^l}{w_{3.2}^u} - 0.09\right| \le \chi \; ; \left|\frac{w_{3.3}^m}{w_{3.2}^m} - 0.33\right| \le \chi \; ; \left|\frac{w_{3.3}^u}{w_{3.2}^l} - 1.34\right| \le \chi; \\[2mm]
\sum_{j=1}^{4}(w_j^l + 4 \cdot w_j^m + w_j^u)/6 = 1 \; , \forall j = 1,2,3,4 \\[2mm]
w_j^l \le w_j^m \le w_j^u, \forall j = 1,2,3,4 \\[1mm]
w_j^l, w_j^m, w_j^u \ge 0 \; , \forall j = 1,2,3,4
\end{cases}
$$

Expert 1　$(C4.1 - C4.2)$ min $\chi$
s.t.

$$
\begin{cases}
\left|\frac{w_{4.2}^l}{w_{4.1}^u} - 2.5\right| \le \chi \; ; \left|\frac{w_{4.2}^m}{w_{4.1}^m} - 3\right| \le \chi \; ; \left|\frac{w_{4.2}^u}{w_{4.1}^l} - 3.5\right| \le \chi \\[2mm]
(w_{2.1}^l + 4 \cdot w_{2.1}^m + w_{2.1}^u)/6 + (w_{2.2}^l + 4 \cdot w_{2.2}^m + w_{2.2}^u)/6 = 1 \\[1mm]
w_j^l \le w_j^m \le w_j^u, \forall j = 1,2 \\[1mm]
w_j^l, w_j^m, w_j^u \ge 0 \; , \forall j = 1,2
\end{cases}
$$

Expert 7　$(C4.1 - C4.2)$ min $\chi$
s.t.

$$
\begin{cases}
\left|\frac{w_{4.1}^l}{w_{4.2}^u} - 2.5\right| \le \chi \; ; \left|\frac{w_{4.1}^m}{w_{4.2}^m} - 3\right| \le \chi \; ; \left|\frac{w_{4.2}^u}{w_{4.1}^l} - 3.5\right| \le \chi \\[2mm]
(w_{2.1}^l + 4 \cdot w_{2.1}^m + w_{2.1}^u)/6 + (w_{2.2}^l + 4 \cdot w_{2.2}^m + w_{2.2}^u)/6 = \\[1mm]
w_j^l \le w_j^m \le w_j^u, \forall j = 1,2 \\[1mm]
w_j^l, w_j^m, w_j^u \ge 0 \; , \forall j = 1,2
\end{cases}
$$

Expert 1　$(C5.1 - C5.3)$ min $\chi$
s.t.

$$
\begin{cases}
\left|\frac{w_{5.1}^l}{w_{5.2}^u} - 0.67\right| \le \chi \; ; \left|\frac{w_{5.1}^m}{w_{5.2}^m} - 1\right| \le \chi \; ; \left|\frac{w_{5.1}^u}{w_{5.2}^l} - 1.5\right| \le \chi; \\[2mm]
\left|\frac{w_{5.2}^l}{w_{5.3}^u} - 2.33\right| \le \chi \; ; \left|\frac{w_{5.2}^m}{w_{5.3}^m} - 4\right| \le \chi \; ; \left|\frac{w_{5.2}^u}{w_{5.3}^l} - 6.7\right| \le \chi; \\[2mm]
\left|\frac{w_{5.1}^l}{w_{5.3}^u} - 1.56\right| \le \chi \; ; \left|\frac{w_{5.1}^m}{w_{5.3}^m} - 4.00\right| \le \chi \; ; \left|\frac{w_{5.1}^u}{w_{5.3}^l} - 10.07\right| \le \chi; \quad \cdots\cdots \\[2mm]
\sum_{j=1}^{3}(w_j^l + 4 \cdot w_j^m + w_j^u)/6 = 1 \; , \forall j = 1,2,3 \\[2mm]
w_j^l \le w_j^m \le w_j^u, \forall j = 1,2,3 \\[1mm]
w_j^l, w_j^m, w_j^u \ge 0 \; , \forall j = 1,2,3
\end{cases}
$$

Expert 7　$(C5.1 - C5.3)$ min $\chi$
s.t.

$$
\begin{cases}
\left|\frac{w_{5.3}^l}{w_{5.2}^u} - 0.67\right| \le \chi \; ; \left|\frac{w_{5.3}^m}{w_{5.2}^m} - 1\right| \le \chi \; ; \left|\frac{w_{5.3}^u}{w_{5.2}^l} - 1.5\right| \le \chi; \\[2mm]
\left|\frac{w_{5.2}^l}{w_{5.1}^u} - 0.67\right| \le \chi \; ; \left|\frac{w_{5.2}^m}{w_{5.1}^m} - 1\right| \le \chi \; ; \left|\frac{w_{5.2}^u}{w_{5.1}^l} - 1.49\right| \le \chi; \\[2mm]
\left|\frac{w_{5.3}^l}{w_{5.1}^u} - 0.45\right| \le \chi \; ; \left|\frac{w_{5.3}^m}{w_{5.1}^m} - 2\right| \le \chi \; ; \left|\frac{w_{5.3}^u}{w_{5.1}^l} - 2.24\right| \le \chi; \\[2mm]
\sum_{j=1}^{3}(w_j^l + 4 \cdot w_j^m + w_j^u)/6 = 1 \; , \forall j = 1,2,3 \\[2mm]
w_j^l \le w_j^m \le w_j^u, \forall j = 1,2,3 \\[1mm]
w_j^l, w_j^m, w_j^u \ge 0 \; , \forall j = 1,2,3
\end{cases}
$$

**Figure A1**

# Appendix C

**Table A2.** Local values of the fuzzy weight coefficients of the sub-criteria.

| | E1-C1-C5 | | | E2-C1-C5 | | | E3-C1-C5 | | | E4-C1-C5 | | | E5-C1-C5 | | | E6-C1-C5 | | | E7-C1-C5 | | | Sum | | |
|---|---|---|---|---|---|---|---|---|---|---|---|---|---|---|---|---|---|---|---|---|---|---|---|---|
| C1 | 0.10 | 0.12 | 0.13 | 0.29 | 0.29 | 0.29 | 0.16 | 0.28 | 0.28 | 0.26 | 0.35 | 0.39 | 0.26 | 0.27 | 0.27 | 0.14 | 0.26 | 0.26 | 0.19 | 0.28 | 0.28 | 0.20 | 0.27 | 0.27 |
| C2 | 0.28 | 0.33 | 0.38 | 0.19 | 0.22 | 0.22 | 0.22 | 0.22 | 0.22 | 0.10 | 0.11 | 0.11 | 0.17 | 0.29 | 0.29 | 0.09 | 0.27 | 0.27 | 0.07 | 0.09 | 0.09 | 0.16 | 0.22 | 0.23 |
| C3 | 0.07 | 0.07 | 0.07 | 0.10 | 0.15 | 0.22 | 0.18 | 0.28 | 0.29 | 0.26 | 0.36 | 0.39 | 0.17 | 0.29 | 0.29 | 0.06 | 0.15 | 0.15 | 0.11 | 0.20 | 0.20 | 0.13 | 0.21 | 0.23 |
| C4 | 0.26 | 0.30 | 0.33 | 0.08 | 0.25 | 0.50 | 0.06 | 0.15 | 0.15 | 0.09 | 0.12 | 0.13 | 0.07 | 0.10 | 0.11 | 0.16 | 0.27 | 0.27 | 0.07 | 0.25 | 0.34 | 0.11 | 0.21 | 0.26 |
| C5 | 0.12 | 0.18 | 0.29 | 0.06 | 0.08 | 0.09 | 0.04 | 0.12 | 0.12 | 0.08 | 0.08 | 0.08 | 0.06 | 0.10 | 0.13 | 0.06 | 0.15 | 0.15 | 0.12 | 0.25 | 0.29 | 0.08 | 0.14 | 0.17 |

| | E1-C1.1-C1.4 | | | E2-C1.1-C1.4 | | | E3-C1.1-C1.4 | | | E4-C1.1-C1.4 | | | E5-C1.1-C1.4 | | | E6-C1.1-C1.4 | | | E7-C1.1-C1.4 | | | | | |
|---|---|---|---|---|---|---|---|---|---|---|---|---|---|---|---|---|---|---|---|---|---|---|---|---|
| C11 | 0.34 | 0.48 | 0.53 | 0.08 | 0.16 | 0.20 | 0.18 | 0.33 | 0.34 | 0.27 | 0.37 | 0.41 | 0.30 | 0.52 | 0.54 | 0.15 | 0.29 | 0.29 | 0.10 | 0.16 | 0.16 | 0.20 | 0.33 | 0.35 |
| C12 | 0.09 | 0.20 | 0.22 | 0.08 | 0.16 | 0.22 | 0.09 | 0.15 | 0.15 | 0.10 | 0.12 | 0.12 | 0.10 | 0.17 | 0.17 | 0.14 | 0.28 | 0.28 | 0.13 | 0.16 | 0.16 | 0.10 | 0.18 | 0.19 |
| C13 | 0.11 | 0.11 | 0.11 | 0.16 | 0.46 | 0.53 | 0.20 | 0.31 | 0.31 | 0.19 | 0.45 | 0.65 | 0.07 | 0.14 | 0.15 | 0.15 | 0.24 | 0.24 | 0.19 | 0.35 | 0.36 | 0.15 | 0.29 | 0.34 |
| C14 | 0.18 | 0.25 | 0.25 | 0.26 | 0.26 | 0.26 | 0.17 | 0.28 | 0.28 | 0.08 | 0.08 | 0.08 | 0.20 | 0.23 | 0.23 | 0.17 | 0.26 | 0.26 | 0.12 | 0.40 | 0.52 | 0.17 | 0.25 | 0.27 |

| | E1-C2.1-C2.2 | | | E2-C2.1-C2.2 | | | E3-C2.1-C2.2 | | | E4-C2.1-C2.2 | | | E5-C2.1-C2.2 | | | E6-C2.1-C2.2 | | | E7-C2.1-C2.2 | | | | | |
|---|---|---|---|---|---|---|---|---|---|---|---|---|---|---|---|---|---|---|---|---|---|---|---|---|
| C21 | 0.73 | 0.73 | 0.85 | 0.46 | 0.46 | 0.69 | 0.46 | 0.46 | 0.69 | 0.46 | 0.46 | 0.69 | 0.46 | 0.46 | 0.69 | 0.50 | 0.67 | 0.83 | 0.46 | 0.46 | 0.69 | 0.51 | 0.53 | 0.73 |
| C22 | 0.24 | 0.24 | 0.29 | 0.46 | 0.46 | 0.69 | 0.46 | 0.46 | 0.69 | 0.46 | 0.46 | 0.69 | 0.46 | 0.46 | 0.69 | 0.33 | 0.33 | 0.33 | 0.46 | 0.46 | 0.69 | 0.41 | 0.41 | 0.58 |

| | E1-C3.1-C3.4 | | | E2-C3.1-C3.4 | | | E3-C3.1-C3.4 | | | E4-C3.1-C3.4 | | | E5-C3.1-C3.4 | | | E6-C3.1-C3.4 | | | E7-C3.1-C3.4 | | | | | |
|---|---|---|---|---|---|---|---|---|---|---|---|---|---|---|---|---|---|---|---|---|---|---|---|---|
| C31 | 0.18 | 0.18 | 0.50 | 0.08 | 0.08 | 0.08 | 0.16 | 0.32 | 0.32 | 0.25 | 0.31 | 0.35 | 0.25 | 0.41 | 0.63 | 0.16 | 0.32 | 0.32 | 0.18 | 0.18 | 0.37 | 0.18 | 0.26 | 0.37 |
| C32 | 0.25 | 0.25 | 0.25 | 0.15 | 0.43 | 0.59 | 0.05 | 0.17 | 0.20 | 0.28 | 0.31 | 0.31 | 0.29 | 0.29 | 0.29 | 0.07 | 0.15 | 0.15 | 0.10 | 0.33 | 0.55 | 0.17 | 0.28 | 0.33 |
| C33 | 0.40 | 0.40 | 0.40 | 0.13 | 0.18 | 0.18 | 0.16 | 0.34 | 0.34 | 0.07 | 0.07 | 0.07 | 0.08 | 0.14 | 0.24 | 0.15 | 0.30 | 0.30 | 0.09 | 0.09 | 0.09 | 0.16 | 0.22 | 0.23 |
| C34 | 0.11 | 0.11 | 0.13 | 0.25 | 0.35 | 0.39 | 0.12 | 0.26 | 0.26 | 0.20 | 0.32 | 0.43 | 0.15 | 0.15 | 0.15 | 0.18 | 0.32 | 0.32 | 0.20 | 0.28 | 0.28 | 0.17 | 0.25 | 0.28 |

| | E1-C4.1-C4.2 | | | E2-C4.1-C4.2 | | | E3-C4.1-C4.2 | | | E4-C4.1-C4.2 | | | E5-C4.1-C4.2 | | | E6-C4.1-C4.2 | | | E7-C4.1-C4.2 | | | | | |
|---|---|---|---|---|---|---|---|---|---|---|---|---|---|---|---|---|---|---|---|---|---|---|---|---|
| C41 | 0.24 | 0.24 | 0.29 | 0.46 | 0.46 | 0.69 | 0.46 | 0.46 | 0.69 | 0.18 | 0.20 | 0.23 | 0.46 | 0.46 | 0.69 | 0.46 | 0.46 | 0.69 | 0.73 | 0.73 | 0.85 | 0.43 | 0.43 | 0.59 |
| C42 | 0.73 | 0.73 | 0.85 | 0.46 | 0.46 | 0.69 | 0.46 | 0.46 | 0.69 | 0.80 | 0.80 | 0.80 | 0.46 | 0.46 | 0.69 | 0.46 | 0.46 | 0.69 | 0.24 | 0.24 | 0.29 | 0.52 | 0.52 | 0.67 |

| | E1-C5.1-C5.3 | | | E2-C5.1-C5.3 | | | E3-C5.1-C5.3 | | | E4-C5.1-C5.3 | | | E5-C5.1-C5.3 | | | E6-C5.1-C5.3 | | | E7-C5.1-C5.3 | | | | | |
|---|---|---|---|---|---|---|---|---|---|---|---|---|---|---|---|---|---|---|---|---|---|---|---|---|
| C51 | 0.29 | 0.44 | 0.65 | 0.18 | 0.18 | 0.18 | 0.33 | 0.33 | 0.33 | 0.50 | 0.57 | 0.63 | 0.43 | 0.57 | 0.71 | 0.49 | 0.49 | 0.49 | 0.23 | 0.23 | 0.56 | 0.35 | 0.40 | 0.51 |
| C52 | 0.44 | 0.44 | 0.44 | 0.45 | 0.54 | 0.63 | 0.22 | 0.33 | 0.49 | 0.14 | 0.14 | 0.14 | 0.29 | 0.29 | 0.29 | 0.15 | 0.15 | 1.11 | 0.30 | 0.30 | 0.41 | 0.28 | 0.31 | 0.50 |
| C53 | 0.06 | 0.11 | 0.19 | 0.18 | 0.27 | 0.42 | 0.33 | 0.33 | 0.33 | 0.20 | 0.28 | 0.42 | 0.10 | 0.14 | 0.20 | 0.19 | 0.19 | 0.26 | 0.33 | 0.38 | 0.53 | 0.20 | 0.24 | 0.34 |

# Appendix D

**Table A3.** Linguistic assessments of potential alternatives against the criteria.

|    |    | C1.1 | C1.2 | C1.3 | C1.4 | C2.1 | C2.2 | C3.1 | C3.2 | C3.3 | C3.4 | C4.1 | C4.2 | C5.1 | C5.2 | C5.3 |
|----|----|------|------|------|------|------|------|------|------|------|------|------|------|------|------|------|
| A1 | E1 | TH | M  | M  | TH | H  | M  | H  | M  | H  | M  | H  | H  | TH | H  | M  |
|    | E2 | TF | TH | F  | TH | M  | TH | TH | TH | TH | TH | TF | TH | TH | TH | TH |
|    | E3 | TH | M  | M  | TH | TH | M  | H  | TF | TH | TH | F  | M  | TH | TH | TF |
|    | E4 | TH | M  | M  | TH | TH | M  | TH | H  | H  | TH | H  | F  | TH | TH | F  |
|    | E5 | TH | M  | F  | TH | H  | M  | TH | H  | H  | TH | M  | TF | TH | H  | F  |
|    | E6 | TH | TH | TH | TH | TH | TH | TH | TH | TH | TH | TH | TH | TH | TH | TH |
|    | E7 | TH | H  | TH | M  | TH | H  | TH | M  | M  | TH | H  | H  | H  | H  | M  |
| A2 | E1 | H  | M  | F  | H  | H  | M  | H  | M  | H  | H  | F  | TF | M  | M  | M  |
|    | E2 | H  | H  | F  | H  | M  | H  | TH | TH | M  | M  | TF | TF | M  | F  | TH |
|    | E3 | TH | M  | F  | TH | TH | M  | H  | TF | H  | M  | TF | F  | TH | M  | M  |
|    | E4 | H  | H  | F  | M  | M  | H  | TH | H  | M  | TH | H  | F  | H  | H  | M  |
|    | E5 | TH | M  | H  | TH | H  | M  | H  | H  | F  | H  | TF | M  | H  | M  | F  |
|    | E6 | F  | F  | F  | F  | M  | F  | M  | M  | M  | M  | M  | M  | F  | F  | F  |
|    | E7 | TH | H  | TH | TH | TH | H  | TH | M  | M  | TH | H  | M  | H  | H  | H  |
| A3 | E1 | F  | M  | F  | M  | H  | M  | H  | M  | M  | M  | F  | H  | M  | M  | H  |
|    | E2 | M  | M  | F  | M  | M  | M  | TH | TH | TH | H  | TF | H  | M  | F  | F  |
|    | E3 | H  | M  | F  | H  | TH | M  | H  | TF | H  | TH | TF | F  | H  | M  | H  |
|    | E4 | TH | TF | M  | F  | H  | TF | TH | H  | TH | H  | M  | F  | H  | H  | TH |
|    | E5 | H  | M  | TF | H  | H  | M  | M  | TH | M  | H  | F  | TF | H  | M  | TH |
|    | E6 | TH | TH | TH | TH | TH | TH | TH | TH | TH | TH | TH | TH | TH | TH | TH |
|    | E7 | TH | H  | TH | TH | TH | H  | TH | M  | M  | TH | H  | M  | H  | H  | TH |
| A4 | E1 | F  | H  | M  | M  | M  | H  | M  | M  | TF | F  | F  | M  | M  | M  | M  |
|    | E2 | M  | M  | F  | TH | M  | M  | TH | TF | TF | TF | TF | TF | M  | F  | H  |
|    | E3 | H  | H  | F  | M  | TH | H  | TH | TF | H  | M  | TF | F  | M  | M  | M  |
|    | E4 | TH | F  | M  | M  | H  | F  | TH | H  | M  | M  | M  | F  | M  | M  | H  |
|    | E5 | H  | M  | M  | TH | H  | M  | M  | H  | M  | H  | F  | TF | M  | F  | H  |
|    | E6 | TH | F  | F  | F  | M  | F  | M  | M  | M  | M  | M  | M  | F  | F  | F  |
|    | E7 | TH | H  | TH | TH | TH | H  | TH | M  | M  | TH | H  | M  | H  | H  | TH |
| A5 | E1 | F  | H  | M  | M  | M  | H  | M  | M  | F  | F  | F  | M  | M  | M  | M  |
|    | E2 | M  | M  | F  | TH | M  | M  | TH | TF | TF | TF | TF | TF | M  | F  | H  |
|    | E3 | H  | H  | F  | F  | TH | H  | TH | TF | H  | M  | TF | F  | F  | M  | M  |
|    | E4 | TH | TH | F  | M  | H  | TH | TH | H  | F  | M  | M  | F  | M  | M  | F  |
|    | E5 | H  | F  | TF | M  | H  | F  | F  | F  | M  | H  | TF | F  | M  | H  | TF |
|    | E6 | TH | F  | F  | F  | M  | F  | M  | M  | M  | M  | M  | M  | F  | F  | F  |
|    | E7 | TH | TH | TH | M  | TH | TH | TH | M  | M  | M  | H  | M  | M  | M  | M  |
| A6 | E1 | TH | F  | TF | TH | H  | F  | H  | M  | H  | F  | F  | F  | H  | F  | F  |
|    | E2 | M  | H  | F  | TH | TF | H  | TH | TF | TF | TF | TF | TF | TH | M  | F  |
|    | E3 | F  | F  | F  | F  | TH | F  | TH | TF | H  | M  | TF | F  | F  | F  | TF |
|    | E4 | H  | M  | F  | F  | H  | M  | TH | H  | M  | F  | M  | F  | M  | M  | M  |
|    | E5 | H  | TF | TF | TH | M  | TF | H  | F  | M  | H  | TF | F  | M  | H  | TF |
|    | E6 | F  | F  | F  | F  | F  | F  | F  | F  | F  | F  | F  | F  | F  | F  | F  |
|    | E7 | H  | TH | TH | M  | TH | TH | TH | M  | M  | M  | H  | H  | M  | TH | H  |
| A7 | E1 | TF | F  | M  | TF | TH | F  | H  | M  | H  | F  | F  | F  | TF | F  | M  |
|    | E2 | H  | H  | F  | TF | M  | H  | TH | TH | H  | M  | TF | TF | TH | TH | M  |
|    | E3 | TH | F  | F  | TH | TH | F  | TH | TF | TH | TH | TF | F  | TH | TH | TH |
|    | E4 | TF | H  | F  | TF | TH | H  | TH | H  | H  | F  | M  | F  | TF | TF | H  |
|    | E5 | F  | F  | TF | TF | M  | F  | M  | F  | F  | H  | TF | F  | H  | TF | TF |
|    | E6 | TH | TH | TH | TH | TH | TH | TH | TH | TH | TH | TH | TH | TH | TH | TH |
|    | E7 | TF | H  | TH | TH | TH | H  | TH | M  | M  | TH | H  | H  | TH | TH | M  |

## Appendix E

**Table A4.** Preference function of alternatives over criteria.

|  |  | C1.1 | C1.2 | C1.3 | C1.4 | C2.1 | C2.2 | C3.1 | C3.2 | C3.3 | C3.4 | C4.1 | C4.2 | C5.1 | C5.2 | C5.3 | $\pi(a,b)$ | | |
|---|---|---|---|---|---|---|---|---|---|---|---|---|---|---|---|---|---|---|---|
| A1 | A2 | 1 | 1 | 1 | 1 | 1 | 0 | 1 | 1 | 1 | 1 | 1 | 1 | 1 | 1 | 0 | 0.45 | 0.90 | 1.26 |
|  | A3 | 1 | 0 | 1 | 1 | 0 | 1 | 1 | 1 | 0 | 1 | 1 | 1 | 1 | 1 | 0 | 0.40 | 0.78 | 1.12 |
|  | A4 | 1 | 0 | 1 | 1 | 1 | 1 | 1 | 1 | 1 | 1 | 1 | 1 | 1 | 1 | 1 | 0.51 | 0.98 | 1.39 |
|  | A5 | 1 | 0 | 1 | 1 | 1 | 1 | 1 | 1 | 0 | 1 | 1 | 1 | 1 | 1 | 1 | 0.49 | 0.93 | 1.34 |
|  | A6 | 1 | 1 | 1 | 1 | 1 | 1 | 1 | 1 | 1 | 1 | 1 | 1 | 1 | 1 | 1 | 0.53 | 1.03 | 1.44 |
|  | A7 | 1 | 1 | 1 | 1 | 0 | 1 | 1 | 1 | 1 | 1 | 1 | 1 | 1 | 1 | 0 | 0.44 | 0.88 | 1.22 |
| A2 | A1 | 0 | 0 | 0 | 0 | 0 | 0 | 0 | 0 | 0 | 0 | 0 | 0 | 0 | 0 | 1 | 0.02 | 0.03 | 0.06 |
|  | A3 | 1 | 0 | 0 | 0 | 0 | 1 | 0 | 1 | 0 | 0 | 0 | 0 | 0 | 0 | 1 | 0.14 | 0.27 | 0.36 |
|  | A4 | 1 | 0 | 0 | 0 | 0 | 1 | 1 | 1 | 1 | 1 | 0 | 0 | 1 | 1 | 1 | 0.26 | 0.52 | 0.73 |
|  | A5 | 1 | 0 | 0 | 0 | 0 | 1 | 1 | 1 | 0 | 1 | 1 | 0 | 0 | 1 | 1 | 0.26 | 0.51 | 0.75 |
|  | A6 | 1 | 1 | 1 | 0 | 1 | 1 | 1 | 1 | 1 | 1 | 0 | 1 | 0 | 1 | 1 | 0.42 | 0.81 | 1.11 |
|  | A7 | 1 | 1 | 1 | 1 | 0 | 1 | 0 | 0 | 1 | 1 | 0 | 0 | 1 | 0 | 1 | 0.28 | 0.56 | 0.70 |
| A3 | A1 | 0 | 1 | 0 | 1 | 0 | 1 | 0 | 0 | 1 | 1 | 0 | 0 | 0 | 0 | 0 | 0.16 | 0.31 | 0.37 |
|  | A2 | 0 | 1 | 0 | 1 | 1 | 0 | 0 | 0 | 1 | 1 | 0 | 1 | 1 | 1 | 0 | 0.29 | 0.54 | 0.75 |
|  | A4 | 1 | 0 | 0 | 1 | 1 | 0 | 1 | 0 | 1 | 1 | 0 | 1 | 1 | 1 | 1 | 0.34 | 0.66 | 0.94 |
|  | A5 | 1 | 0 | 0 | 1 | 1 | 0 | 1 | 0 | 1 | 1 | 1 | 0 | 1 | 1 | 1 | 0.33 | 0.65 | 0.92 |
|  | A6 | 1 | 1 | 1 | 1 | 1 | 1 | 1 | 1 | 1 | 1 | 1 | 0 | 1 | 1 | 1 | 0.47 | 0.92 | 1.27 |
|  | A7 | 1 | 1 | 1 | 1 | 0 | 1 | 0 | 0 | 1 | 1 | 0 | 0 | 1 | 1 | 0 | 0.28 | 0.57 | 0.73 |
| A4 | A1 | 0 | 1 | 1 | 0 | 0 | 1 | 0 | 0 | 1 | 0 | 0 | 0 | 0 | 0 | 0 | 0.14 | 0.26 | 0.33 |
|  | A2 | 0 | 1 | 1 | 1 | 0 | 0 | 0 | 0 | 0 | 0 | 0 | 0 | 0 | 0 | 0 | 0.08 | 0.19 | 0.22 |
|  | A3 | 0 | 0 | 1 | 0 | 0 | 0 | 0 | 1 | 0 | 0 | 0 | 0 | 0 | 0 | 0 | 0.05 | 0.14 | 0.17 |
|  | A5 | 1 | 0 | 1 | 0 | 0 | 0 | 1 | 0 | 0 | 0 | 1 | 0 | 0 | 0 | 0 | 0.14 | 0.31 | 0.43 |
|  | A6 | 0 | 1 | 1 | 0 | 1 | 1 | 0 | 1 | 1 | 1 | 1 | 0 | 0 | 0 | 0 | 0.31 | 0.58 | 0.79 |
|  | A7 | 1 | 1 | 1 | 1 | 0 | 1 | 0 | 0 | 0 | 1 | 0 | 0 | 1 | 0 | 0 | 0.24 | 0.48 | 0.59 |
| A5 | A1 | 0 | 1 | 0 | 0 | 0 | 0 | 0 | 0 | 0 | 0 | 0 | 0 | 0 | 0 | 0 | 0.02 | 0.05 | 0.05 |
|  | A2 | 0 | 1 | 0 | 1 | 0 | 0 | 0 | 0 | 1 | 0 | 0 | 1 | 0 | 0 | 0 | 0.13 | 0.27 | 0.35 |
|  | A3 | 0 | 1 | 1 | 1 | 1 | 1 | 1 | 1 | 1 | 0 | 1 | 1 | 1 | 1 | 0 | 0.45 | 0.85 | 1.23 |
|  | A4 | 0 | 1 | 0 | 1 | 0 | 1 | 0 | 1 | 1 | 1 | 0 | 1 | 1 | 1 | 0 | 0.29 | 0.57 | 0.79 |
|  | A6 | 0 | 1 | 1 | 1 | 1 | 1 | 0 | 1 | 1 | 1 | 0 | 0 | 1 | 0 | 0 | 0.32 | 0.61 | 0.79 |
|  | A7 | 1 | 1 | 1 | 1 | 0 | 1 | 0 | 0 | 1 | 1 | 0 | 0 | 1 | 0 | 0 | 0.26 | 0.53 | 0.65 |
| A6 | A1 | 0 | 0 | 0 | 0 | 0 | 0 | 0 | 0 | 0 | 0 | 0 | 0 | 0 | 0 | 0 | 0.00 | 0.00 | 0.00 |
|  | A2 | 0 | 0 | 0 | 1 | 0 | 0 | 0 | 0 | 0 | 0 | 0 | 1 | 0 | 0 | 0 | 0.09 | 0.17 | 0.25 |
|  | A3 | 0 | 0 | 0 | 0 | 0 | 0 | 0 | 0 | 0 | 0 | 0 | 0 | 0 | 0 | 0 | 0.00 | 0.00 | 0.00 |
|  | A4 | 1 | 0 | 0 | 0 | 0 | 0 | 0 | 0 | 0 | 0 | 0 | 1 | 1 | 1 | 1 | 0.16 | 0.32 | 0.49 |
|  | A6 | 1 | 0 | 0 | 0 | 0 | 0 | 1 | 0 | 0 | 0 | 1 | 0 | 0 | 1 | 1 | 0.15 | 0.31 | 0.47 |
|  | A7 | 1 | 0 | 1 | 1 | 0 | 1 | 0 | 0 | 0 | 1 | 0 | 0 | 1 | 0 | 0 | 0.22 | 0.43 | 0.54 |
| A7 | A1 | 0 | 0 | 0 | 0 | 0 | 0 | 0 | 0 | 0 | 0 | 0 | 0 | 0 | 0 | 1 | 0.02 | 0.03 | 0.06 |
|  | A2 | 0 | 0 | 0 | 0 | 1 | 0 | 1 | 0 | 0 | 0 | 1 | 1 | 0 | 1 | 0 | 0.23 | 0.41 | 0.66 |
|  | A3 | 0 | 0 | 0 | 0 | 0 | 0 | 1 | 1 | 0 | 0 | 1 | 1 | 0 | 0 | 0 | 0.15 | 0.31 | 0.49 |
|  | A4 | 0 | 0 | 0 | 0 | 1 | 0 | 1 | 1 | 1 | 0 | 1 | 1 | 0 | 1 | 1 | 0.29 | 0.55 | 0.85 |
|  | A5 | 0 | 0 | 0 | 0 | 1 | 0 | 1 | 1 | 0 | 0 | 1 | 1 | 0 | 1 | 1 | 0.27 | 0.50 | 0.80 |
|  | A7 | 0 | 1 | 0 | 0 | 1 | 0 | 1 | 1 | 0 | 0 | 1 | 1 | 0 | 1 | 1 | 0.29 | 0.55 | 0.85 |

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
