# Peer review of "Novel Fuzzy Composite Indicators for Locating a Logistics Platform under Sustainability Perspectives"

_sustainability, doi:10.3390/su13073891_

Round 1

Reviewer 1 Report

This paper presents an application of multi criteria decision making techniques to build composite indicators to assess the viability and sustainability of several possible locations of a logistic platform. While real applications of this type of methodologies in the sustainability context are always interesting, I feel that the manuscript cannot be accepted for publication due to the following reasons:

1.- From the formal point of view, the paper is written in a very careless way, and it is very hard to follow it. Apart from the English of the paper, which needs a revision, there are many other problems with the presentation. Let me point out some examples:

  • There are notation problems here and there. k is used several times to indicate different things (rank in (1), experts later on…); the authors confuse indices, like n, m, k, with the number of elements of the index set (for example, j(k) in (1) should be j(n)). There are plenty of cases like this.
  • Some variables are used without being previously defined, like the elements of (2).
  • In expression (6), for example (there are others), what is the meaning of “for all j”, when j does not appear in the corresponding expression (like the two first constraints) or when it is just the summation index (like the third constraint).
  • In page 12, it is said that “The comparison is made with respect to the first -ranked (most significant) criterion”, but this is not coherent with the evaluations shown in Table 5. Besides, if evaluations are made in a comparative way, then the absolute linguistic terms defined in table 4 make no sense.
  • Some references to tables are wrong, like the reference to Table 8 in page 12.
  • There are words in French in several places in the text.
  • Etc.

2.- From the methodological point of view, the election of the methods is not justified at all. The two aggregation methods chosen are very different in nature, and they imply different assumptions on many aspects (not only the compensatory issue). Therefore, the joint use of both methods is not justified. Besides, what if they had produced different rankings? How should the decision be made in that case? I also disagree with the non-compensatory character of the PROMETHEE method. If the net flow is used, the method is, at most, partially compensatory. Besides, the authors use the usual preference function without a clear justification.

3.- Regarding the construction of the system of indicators, some of them seem hard to assess, and moreover, one should think that they do not depend of the location chosen, like the fiscal policy, the air pollution, the noise, the current policy or the support role for industry. How do these indicators differ from one location to another?

4.- The sensitivity analysis is weak. The normalization scheme used is not the only source of instability. Besides, it is obvious that this does not affect the results in PROMETHEE. In fact, a prior normalization is not needed in this case, because it is the preference functions that act as normalizers. If the usual function is used, then it is obvious that the results will be the same with any prior normalization, but what if other preference functions are used. A more in-depth robustness analysis should be carried out on weights, evaluations, etc.

5.- Finally, the literature review ignores a field that is crucial to this paper: how are composite indicators built in the literature using multi criteria decision making methods? There are many publications in this field and relating these composite indicators to the assessment of sustainability. For example, Saisana and Tarantola (2002), Nardo et al. (2005) and Munda (2008) contain reviews of methodologies and issues to be taken into account. They also discuss the compensability issue, which is particularly addressed in Munda and Nardo (2009) and El Gibari et al. (2019). Besides, there are methodologies which explicitly allow the construction of composite indicators with different compensation degrees, like Blancas et al. (2010), Taburasi and Guarini (2013), or Ruiz et al. (2020). I think it would be more logical to use these types of methods to compare compensatory and non-compensatory results, based on a single philosophy to build composite indicators, rather than using two completely different approaches. Finally, there are also methodologies designed to build partially compensatory composite indicators, like García-Bernabéu et al. (2020), Mazziota and Pareto (2020) or Ruiz and Cabello (2021).

Author Response

Dear Reviewer,

Please see the attachment,

Best regards,

Hana Ayadi and co-authors

Reviewer 2 Report

The authors presented an interesting study proving the need to include two additional factors (political and territorial) in the criteria for the location of transport and logistics infrastructure facilities. The manuscript contains a detailed description of the proposed methodology and case study for the Tunisian conditions.

1. I suggest moving the list of criteria for the case study (table 3) to section 3.1. The selected criteria are universal and not specific to the case study presented. In addition, the choice of criteria is sufficiently convincingly substantiated by the authors in section 3.1 "Definition of criteria and alternatives".

2. Please check the numbering of tables and figures: Tables with numbers 1, 3, 4, 7 are present twice in the text; Table number 2 is located after Table number 4; Figures with number 1 are present twice.

3. Correct the layout of tables # 4 (Site evaluation criteria) and # 2 (Table 2: The five normalization methods). The text in some columns is unreadable, the brackets are shifted to the next line.

4. Formula 19 is not numbered.

5. It is rational to transfer some bulky tables to the application, for example Table 15.

Author Response

(The authors gave the same response as above.)

Reviewer 3 Report

Dear authors,

I have a few comments on the article:

  • Chapter 2 need to be connected with text. Also, you need to make introduction before you start chapter 2.1.
  • Line 555. There is mistake with the dot.
  • Line 619-622, 774-785, 941-958. You need to correct the lines according to the template. They are not aligned properly.
  • Line 679. It should be written “We chose”.
  • Table 2. Text “method” isn’t clear to read. There is also some text on table 4 that you can’t read.
  • Name of the table 4 need to be connected with text.
  • Lin 802. There is too much space in text.
  • Heading of the table should be repeated on every page if the table is split on multiple pages.
  • Figure 2 has some text on French.
  • You should remove numbers (0,2,4,6,8) on figure 2 and 3. It is not necessary.
  • Remove legend on the figure 4.
  • Tables and figures are numbered wrong. Some figures and tables have the same numeration. Also, the tables are missing some numeration and they are not numbered according to the appearance in the text.
  • What is the difference between criteria C11 I C51? The are very similar. You need to explain the difference.
  • On figure 1: Schematic representation of the proposed approach you have Phase 1: Definition of criteria and alternatives. Does this mean that for every case study you need to make different criteria? Is it possible to make criteria that will be the same for all case studies? Also, make the framework of your model.
  • How do you define what mode of transportation is better? You should explain that criteria in more detail. In you case study, all location has an access to the port. What would happened if one of them doesn’t have it? What about a river transportation? I think that your model is done for your case study and it would be hard to implement it for other problems.

Best regards, reviewer

Author Response

(The authors gave the same response as above.)

Reviewer 4 Report

  1. I would suggest the author apply these chapters to organize this paper. Chapter 1 Introduction; Chapter 2 Literature Review; Chapter 3 Method; Chapter 4 Results; Chapter 5 Implications; Chapter 6 Conclusions; and References. In the present version, I think this might confuse the readers.
  2. The three main criteria for this manuscript are: (a) quality and content of the research/review; (b) Quality, brevity and clarity of presentation; (c) Significance, relevance and timeliness of the topic. In addition, this title is (i) coverage of the literature/significant developments in the field or clarity of discussion within an emerging topic; (ii) originality, new perspectives or insights; (iii) international interest; and (iv) relevance for governance, policy or practical perspectives relevant to the focus of this manuscript. However, this study is lacking the most essential criteria. Hence, I think the author needs to consider these criteria before your submission.
  3. The manuscript is hard to follow, using too many abbreviations. Indeed, the full terms and the abbreviations are repeating during the context.
  4. Please make sure that a competent editor checks the English. Use of the first person (“I,” “we,” etc.) and third-person ("he," "she" etc.) must be avoided.
  5. You have to re-write your paper completely. The abstract should state briefly the purpose of the research, the principal results and major conclusions. An abstract is often presented separately from the article, so it must be able to stand alone.
  6. Please underscore the scientific value-added of your paper in your abstract and introduction.
  7. The introduction should be clearly stated research questions and targets first. Then answer several questions: Why is the topic important (or why do you study on it)? What are research questions? What has been studied? What are your contributions? Why is to propose this particular method? An outline of the paper can also be included. Please build upon the great work we have published on these subjects.
  8. Especially, the introduction section needs to re-organize. The major debate or Argument is not clearly stated in the introduction session. Hence, the contribution debates are weak in this manuscript. I would suggest the author enhance your literature discussion and arrives at your debate or argument.
  9. The mathematical formulation is logically and clearly presented. In addition, the case and associated data analysis are illustrative to demonstrate the usefulness of the proposed method.
  10. Table 4 needs to rearrange, it is not clear in the current version.
  11. Please explain the tables in more detail and interpreted what those tables presented.
  12. Please explain your results in steps and links to your proposed method.
  13. I would like to request the author to emphasize the contributions practically and academically in the implications session.
  14. Please put particular emphasis on its novelty and expected significance for the field of environmental science and technology or MCDM field.
  15. Please make sure your conclusions section underscore the scientific value-added of your paper, and/or the applicability of your findings/results, as indicated previously. Please revise your conclusion part into more details.  Basically, you should enhance your contributions, limitations, underscore the scientific value-added of your paper, and/or the applicability of your findings/results and future study in this session.

Author Response

(The authors gave the same response as above.)

Round 2

Reviewer 1 Report

The authors have addressed some of my concerns in this revised version, but it is not enough.

From the formal point of view, I still see many flaws in the paper, which is still careless. We can find, for example, “for all j” in an expression where j is the summation index, or (lines 620-621) A11(k), and then m, n and t… But besides there are many format issues with the paper not to mention the layout of the equation of Step 2 (page 16), or the blank spaces in pages 29-30… This is not a proper way to present a paper for revision.

These aspects can obviously be fixed, but I am not convinced about the authors responses to some critical issues that I brought up in my previous report.

  1. From the methodological point of view, I am not convinced by the election of the methods. I do not think that simplicity is an important aspect in a scientific journal and besides, other methods are equally (or more) simple. My point is that the methods chosen are of very different nature. Therefore, the possible differences in the scores obtained are not necessarily due to the compensation degree, but to a whole bunch of reasons. In this particular case, they produce the same ranking, but if the authors are proposing a general methodology for these location problems, a procedure indicating that to do if this does not happen is vital. I understand that the authors are not comparing the methods, but what is the aim of using both of them? The joint use of two methods should provide some more information to the users than just applying one. Otherwise, it makes no sense.
  2. I still have reservations about the use of PROMETHEE as done in the paper Again, I insist that if the net flow is used, PROMETHEE is, at most, partially compensatory. If not, there are incomparabilities that are not shown in the final solution. And again, I am not convinced by the choice of the usual preference function. It means that the slightest difference in favor of one alternative means considering the whole weight of the criterion when building the flows. I think this does not correspond to a fuzzy environment.
  3. The authors have explained some of the criteria, but I still do not understand how can “Fiscal Policy”, “Current Policy” or “Support role for industry” vary from one location to another in the same city.
  4. With respect to the sensitivity analysis, I insist that it does not make sense to carry out the sensitivity analysis on the normalization scheme for PROMETHEE. On the one hand, because PROMETHEE does not need criteria to be normalized (moreover, it is better to use their original measures to establish the thresholds). On the other hand, because the use of the usual preference function obviously produces the same results for any normalization.
  5. Finally, the choice of the triangular numbers defining the fuzzy assessments is also arbitrary, and in principle, it does not seem to respond to the experts’ uncertainty. The effect of these numbers in the final scores has not been studied.

Author Response

(The authors gave the same response as above.)

Reviewer 3 Report

Dear authors,

You have changed all my recommendation for the paper. Best

Best regards, reviewer

Author Response

Dear Reviewer,

We would like to thank you for your careful assessments and constructive comments on our manuscript, particularly the time being spent.

Best regards,

Hana Ayadi and co-authors

Reviewer 4 Report

  1. I would suggest the author apply these chapters to organize this paper. Chapter 1 Introduction; Chapter 2 Literature Review; Chapter 3 Method; Chapter 4 Results; Chapter 5 Implications; Chapter 6 Conclusions; and References. In the present version, I think this might confuse the readers.
  2. The three main criteria for this manuscript are (a) quality and content of the research/review; (b) Quality, brevity and clarity of presentation; (c) Significance, relevance and timeliness of the topic. In addition, this title is (i) coverage of the literature/significant developments in the field or clarity of discussion within an emerging topic; (ii) originality, new perspectives or insights; (iii) international interest; and (iv) relevance for governance, policy or practical perspectives relevant to the focus of this manuscript. However, this study is lacking the most essential criteria. Hence, I think the author needs to consider these criteria before your submission.
  3. The manuscript is hard to follow, using too many abbreviations. Indeed, the full terms and the abbreviations are repeating during the context.
  4. Please make sure that a competent editor checks the English. Use of the first person (“I,” “we,” etc.) and third-person ("he," "she" etc.) must be avoided.
  5. The abstract should state briefly the purpose of the research, the principal results and major conclusions. An abstract is often presented separately from the article, so it must be able to stand alone. 
  6. Please underscore the scientific value-added of your paper in your abstract and introduction.
  7. The introduction should be clearly stated research questions and targets first. Then answer several questions: Why is the topic important (or why do you study on it)? What are research questions? What has been studied? What are your contributions? Why is it propose this particular method? An outline of the paper can also be included. Please build upon the great work we have published on these subjects.
  8. The major defect of this study is the debate or Argument is not clearly stated in the introduction session. Hence, the contribution is weak in this manuscript. I would suggest the author enhance your theoretical discussion and arrives at your debate or argument.
  9. The justification for the criteria Table is insufficient. This is a major oversight given the importance of the table to the paper. It is still not completely clear how these criteria were identified.
  10. Please explain the tables in more detail and interpreted what those tables presented.
  11. Please explain your results in steps and links to your proposed method.
  12. I would like to request the author to emphasize the contributions practically and academically in the implications session.
  13. Please make sure your conclusions section underscore the scientific value-added of your paper, and/or the applicability of your findings/results, as indicated previously. Please revise your conclusion part into more details.  Basically, you should enhance your contributions, limitations, underscore the scientific value-added of your paper, and/or the applicability of your findings/results and future study in this session.

Author Response

Dear Reviewer,

Please see the file in attachment,

Best regards,

Hana Ayadi and co-authors

Round 3

Reviewer 1 Report

After two revisions, I am afraid that the authors have not convinced me about my two main reservations.

  1. I still think that the choice of the methodologies used is not justified. As I previously mentioned, there are two issues with this. First, they are completely different methodologies, with different underlying philosophies. Therefore, they produce different results not only because of their different compensation degrees, but because of the differences in the whole procedure. Second, what is the aim of using both? In my opinion, the issue is not just providing rankings, but providing useful and significant information and feedback about the problem. Therefore, what does each method add with respect to the other one. If different rankings were produced, what are the different results telling us. It only makes sense to use both methodologies if this issue is addressed. Otherwise, the results are not giving us significant information.
  2. I still do not agree with the authors’ justification about the use of the usual preference function for all the criteria in PROMETHEE. Not having to provide thresholds is not a justification. For example, you can use the linear criterion, without the indifference threshold and with the maximum difference as p. This allows to take into account the differences in the values of the criteria, which seems logical. In my opinion, using the usual preference function for all the criteria actually undermines the potential usefulness of PROMETHEE.

Reviewer 4 Report

The authors have been addressed the comments from reviewers; thus, I like to nominate this study to publish in Sustainability.